# Meta-Referential Games to Learn Compositional Learning Behaviours

## Abstract

Human beings use compositionality to generalise from past experiences to novel experiences. We assume a separation of our experiences into fundamental atomic components that can be recombined in novel ways to support our ability to engage with novel experiences. We frame this as the ability to learn to generalise compositionally, and we will refer to behaviours making use of this ability as compositional learning behaviours (CLBs). A central problem to learning CLBs is the resolution of a binding problem (BP). While it is another feat of intelligence that human beings perform with ease, it is not the case for state-of-the-art artificial agents. Thus, in order to build artificial agents able to collaborate with human beings, we propose to develop a novel benchmark to investigate agents' abilities to exhibit CLBs by solving a domain-agnostic version of the BP. We take inspiration from the language emergence and grounding framework of referential games and propose a meta-learning extension of referential games, entitled Meta-Referential Games, and use this framework to build our benchmark, the Symbolic Behaviour Benchmark (S2B). We provide baseline results and error analysis showing that our benchmark is a compelling challenge that we hope will spur the research community towards developing more capable artificial agents.

## 1 Introduction

Defining compositional behaviours (CBs) as "the ability to generalise from combinations of **trained-on** atomic components to novel re-combinations of those very same components", then we can define compositional learning behaviours (CLBs) as "the ability to generalise **in an online fashion** from a few combinations of never-before-seen atomic components to novel re-combinations of those very same components". We employ the term online here to imply a few-shot learning context (Vinyals et al., 2016; Mishra et al., 2018) that demands that artificial agents learn from, and then leverage some novel information, both over the course of a single lifespan, or episode, in our case of few-shot meta-RL (see Beck et al. (2023) for a review of meta-RL). Thus, CLBs involve a few-shot learning aspect that is not present in CBs. In this paper, we investigate artificial agents' abilities for CLBs.

**Compositional Learning Behaviours as Symbolic Behaviours.** Santoro et al. (2021) states that a symbolic entity does not exist in an objective sense but solely in relation to an "*interpreter who treats it as such*", and thus the authors argue that there exists a set of behaviours, i.e. *symbolic behaviours*, that are consequences of agents engaging with symbols. Thus, in order to evaluate artificial agents in terms of their ability to collaborate with humans, we can use the presence or absence of symbolic behaviours. The authors propose to characterise symbolic behaviours as receptive, constructive, embedded, malleable, separable, meaningful, and graded. In this work, we will primarily focus on the receptivity and constructivity aspects. Receptivity aspects amount to the fact that the agents should be able to receive new symbolic conventions in an online fashion. For instance, when a child introduces an adult to their toys' names, the adults are able to discriminate between those new names upon the next usage. Constructivity aspects amount to the fact that the agents should be able to form new symbolic conventions in an online fashion. For instance, when facing novel situations that require collaborations, two human teammates can come up with novel referring expressions to easily discriminate between different events occurring. Both aspects reflect to abilities that support collaboration. Thus, we develop a benchmark that tests the abilities of artificial agents to perform receptive and constructive behaviours, and the current paper will primarily focus on testing for CLBs.

**Binding Problem & Meta-Learning.** Following Greff et al. (2020), we refer to the binding problem (BP) as the "inability to dynamically and flexibly bind[/re-use] information that is distributed throughout the [architecture]" of some artificial agents (modelled with artificial neural networks here). We note that there is an inherent BP that requires solving for agents to exhibit CLBs. Indeed, over the course of a single episode (as opposed to a whole training process, in the case of CBs), agents must dynamically identify/segregate the component values from the observation of multiple stimuli, timestep after timestep, and then (re-)bind/(re-)use/(re-)combine this information (hopefully stored in some memory component of their architecture) in order to respond correctly when novel stimuli are presented to them. Solving the BP instantiated in such a context, i.e. re-using previously-acquired information in ways that serve the current situation, is another feat of intelligence that human beings perform with ease, on the contrary to current state-of-the-art artificial agents. Thus, our benchmark must emphasise testing agents' abilities to exhibit CLBs by solving a version of the BP. Moreover, we argue for a domain-agnostic BP, i.e. not grounded in a specific modality such as vision or audio, as doing so would limit the external validity of the test. We aim for as few assumptions as possible to be made about the nature of the BP we instantiate (Chollet, 2019). This is crucial to motivate the form of the stimuli we employ, and we will further detail this in Section 3.1.

**Language Grounding & Emergence.** In order to test the quality of some symbolic behaviours, our proposed benchmark needs to query the semantics that agents (*the interpreters*) may extract from their experience, and it must be able to do so in a referential fashion (e.g. being able to query to what extent a given experience is referred to as, for instance, 'the sight of a red tomato'), similarly to most language grounding benchmarks. Subsequently, acknowledging that the simplest form of collaboration is maybe the exchange of information, i.e. communication, via a given code, or language, we argue that the benchmark must therefore also allow agents to manipulate this code/language that they use to communicate. This property is known as the metalinguistic/reflexive function of languages (Jakobson, 1960). It is mainly investigated in the current deep learning era within the field of language emergence( Lazaridou and Baroni (2020), and see Brandizzi (2023) and Denamganaï and Walker (2020a) for further reviews), via the use of variants of the referential games (RGs) (Lewis, 1969). Thus, we take inspiration from the language emergence and grounding framework of RGs, where (i) the language domain represents a semantic domain that can be probed and queried, and (ii) the reflexive function of language is indeed addressed. Then, in order to instantiate different BPs at each episode (preventing the agent from 'cheating' by combining experiences from one episode to another), we propose a meta-learning extension to RGs, entitled Meta-Referential Games, and use this framework to build our benchmark. The resulting multi-agent/single-agent benchmark, entitled Symbolic Behaviour Benchmark (S2B), has the potential to test for more symbolic behaviours pertaining to language grounding and emergence rather than solely CLBs. In the present paper, though, we solely focus on the language grounding task of learning CLBs. After review of the background (Section 2) , we will present our contributions as follows:

- We propose the Symbolic Behaviour Benchmark to enables evaluation of symbolic behaviours (Section 3), and provide a detailed investigation in the case of CLBs.
- We present a novel stimulus representation scheme, entitled Symbolic Continuous Stimulus (SCS) representation which is able to instantiate a BP, on the contrary to more common symbolic representations like the one/multi-hot encoded scheme (Section 3.1);
- We propose the Meta-Referential Game framework, a meta-learning extension to common RGs, that is built over SCS-represented stimuli (Section 3.2);
- We provide baseline results for both the multi-agent and single-agent contexts, along with evaluations of the performance of the agents in terms of CLBs and linguistic compositionality of the emerging languages, showing that our benchmark is a compelling challenge that we hope will spur the research community (Section 4).

## 2  BACKGROUND

The first instance of an environment that demonstrated a primary focus on the objective of communicating efficiently is the *signaling game* or *referential game* (RG) by Lewis (1969), where a speaker agent is asked to send a message to the listener agent, based on the *state/stimulus* of the world that it observed. The listener agent then acts upon the observation of the message by choosing one of the *actions* available to it. Both players' goals are aligned (it features *pure coordination/common*

*interests*), with the aim of performing the 'best' *action* given the observed *state*.In the recent deep learning era, many variants of the RG have appeared. Following the nomenclature proposed in Denamganaï and Walker (2020b), Figure 1 illustrates in the general case a *discriminative* 2-*players / L-signal / N-round / K-distractors / descriptive / object-centric* variant. In short, the speaker receives a stimulus and communicates with the listener (up to $N$ back-and-forth using messages of at most $L$ tokens each), who additionally receives a set of $K+1$ stimuli (potentially including a semantically-similar stimulus as the speaker, referred to as an object-centric stimulus). The task is for the listener to determine, given communication with the speaker, whether any of its observed stimuli match the speaker's. Among the many possible features found in the literature, we highlight here those that will be relevant to how S2B is built, and then provide formalism that we will abide by throughout the paper. The **number of communication rounds** $N$ characterises (i) whether the listener agent can send messages back to the speaker agent and (ii) how many communication rounds can be expected before the listener agent is finally tasked to decide on an action. The basic (discriminative) *RG* is **stimulus-centric**, which assumes that both agents would be somehow embodied in the same body, and they are tasked to discriminate between given stimuli, that are the results of one single perception 'system'. On the other hand, Choi et al. (2018) introduced an **object-centric** variant which incorporates the issues that stem from the difference of embodiment (which has been later re-introduced under the name *Concept game* by Mu and Goodman (2021)). The agents are tasked with discriminating between objects (or scenes) independently of the viewpoint from which they may experience them. In the object-centric variant, the game is more about bridging the gap between each other's cognition rather than (just) finding a common language. The adjective 'object-centric' is used to qualify a stimulus that is different from another but actually present the same meaning (e.g. same object, but seen under a different viewpoint).

Following the last communication round, the listener outputs a decision ($D_i^L$ in Figure 2) about whether any of the stimulus it is observing matches the one (or a semantically similar one, in object-centric RGs) experienced by the speaker, and if so its action index must represent the index of the stimulus it identifies as being the same. In the case of a **descriptive** variant, it may as well be possible that none of the stimuli are the same as the target one, therefore the action of index 0 is required for success. We talk about accuracy of the listener agent to refer to its ability to make the correct decision over multiple RGs.

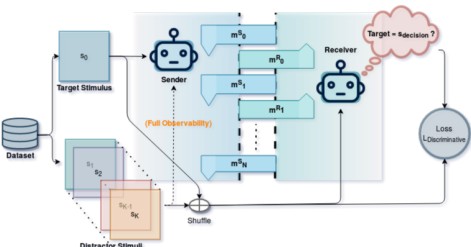

Figure 1: Illustration of a *discriminative* 2-*players / L-signal / N-round* variant of a *RG*.

**Compositionality, Disentanglement & Systematicity.** Compositionality is a phenomenon that human beings are able to identify and leverage thanks to the assumption that reality can be decomposed over a set of "disentangle[d,] underlying factors of variations" (Bengio, 2012), and our experience is a noisy, entangled translation of this factorised reality. This assumption is critical to the field of unsupervised learning of disentangled representations (Locatello et al., 2020) that aims to find "manifold learning algorithms" (Bengio, 2012), such as variational autoencoders (VAEs (Kingma and Welling, 2013)), with the particularity that the latent encoding space would consist of disentangled latent variables (see Higgins et al. (2018) for a formal definition). As a concept, compositionality has been the focus of many definition attempts. For instance, it can be defined as "the algebraic capacity to understand and produce novel combinations from known components"(Loula et al. (2018) referring to Montague (1970)) or as the property according to which "the meaning of a complex expression is a function of the meaning of its immediate syntactic parts and the way in which they are combined" (Krifka, 2001). Although difficult to define, the commmunity seems to agree on the fact that it would enable learning agents to exhibit systematic generalisation abilities (also referred to as combinatorial generalisation (Battaglia et al., 2018)). Some of the ambiguities that come with those loose definitions start to be better understood and explained, as in the work of Hupkes et al. (2019). While often studied in relation to languages, it is usually defined with a focus on behaviours. In this paper, we will refer to linguistic compositionality when considering languages, and interchangeably compositional behaviours and systematicity to refer to "the ability to entertain a given thought implies the ability to entertain thoughts with semantically related contents"(Fodor et al., 1988).

Linguistic compositionality can be difficult to measure. Brighton and Kirby (2006)'s *topographic similarity* (**topsim**) which is acknowledged by the research community as the main quantitative

metric (Lazaridou et al., 2018; Guo et al., 2019; Słowik et al., 2020; Chaabouni et al., 2020; Ren et al., 2020). Recently, taking inspiration from disentanglement metrics, Chaabouni et al. (2020) proposed two new metrics entitled **posdis** (positional disentanglement metric) and **bosdis** (bag-of-symbols disentanglement metric), that have been shown to be differently 'opinionated' when it comes to what kind of linguistic compositionality they capture. As hinted at by Choi et al. (2018); Chaabouni et al. (2020) and Dessi et al. (2021), linguistic compositionality and disentanglement appears to be two sides of the same coin, in as much as emerging languages are discrete and sequentially-constrained unsupervisedly-learned representations. In Section 3.1, we further develop this bridge between the fields of (compositional) language emergence and unsupervised learning of disentangled representations by asking *what would an ideally-disentangled latent space look like?* and investigate how to leverage it in building our proposed benchmark.

**Richness of the Stimuli & Systematicity.** Chaabouni et al. (2020) found that linguistic compositionality is not necessary to bring about systematicity, as shown by the fact that non-compositional languages wielded by symbolic (generative) RG players were enough to support success in zero-shot compositional tests (ZSCTs). They found that the emergence of a (posdis)-compositional language was a sufficient condition for systematicity to emerge. Finally, they found a necessary condition to foster systematicity, that we will refer to as richness of stimuli condition (Chaa-RSC). It was framed as (i) having a large stimulus space $|I| = i_{val}^{i_{attr}}$, where $i_{attr}$ is the number of attributes/factor dimensions, and $i_{val}$ is the number of possible values on each attribute/factor dimension, and (ii) making sure that it is densely sampled during training, in order to guarantee that different values on different factor dimensions have been experienced together. In a similar fashion, Hill et al. (2019) also propose a richness of stimuli condition (Hill-RSC) that was framed as a data augmentation-like regularizer caused by the egocentric viewpoint of the studied embodied agent. In effect, the diversity of viewpoint allowing the embodied agent to observe over many perspectives the same and unique semantical meaning allows a form of contrastive learning that promotes the agent's systematicity.

## 3 SYMBOLIC BEHAVIOUR BENCHMARK

The version of the S2B that we present in this paper is focused on evaluating receptive and constructive behaviour traits via a single task built around 2-players multi-agent RL (MARL) episodes where players engage in a series of RGs (cf. lines 11 and 17 in Alg. 5 calling Alg. 3). We denote one such episode as a meta-RG and detail it in Section 3.2. Each RG within an episode consists of $N + 2$ RL steps, where $N$ is the *number of communication rounds* available to the agents (cf. Section 2). At each RL step, agents both observe similar or different *object-centric* stimuli and act simultaneously from different actions spaces, depending on their role as the speaker or the listener of the game. Stimuli are presented to the agent using the Symbolic Continuous Stimulus (SCS) representation that we present in Section 3.1. Each RG in a meta-RG follows the formalism laid out in Section 2, with the exception that speaker and listener agents speak simultaneously and observe each other's messages upon the next RL step. Thus, at step $N + 1$, the speaker's action space consists solely of a *no-operation* (NO-OP) action while the listener's action space consists solely of the decision-related action space. In practice, the environment simply ignores actions that are not allowed depending on the RL step. Next, step $N + 2$ is intended to provide feedback to the listener agent as its observation is replaced with the speaker's observation (cf. line 12 and 18 in Alg. 5). Note that this is the exact stimulus that the speaker has been observing, rather than a **possible** object-centric sample. We will investigate how the nature of the feedback impacts emergent languages in a subsequent work. In Figure 3, we present SCS-represented stimuli, as observed by a speaker agent over the course of a typical episode.

### 3.1 SYMBOLIC CONTINUOUS STIMULUS REPRESENTATION

Building about successes of the field of unsupervised learning of disentangled representations, to the question *what would an ideally-disentangled latent space look like?*, we propose the Symbolic Continuous Stimulus (SCS) representation and provide numerical evidence of it in Appendix D.2. It is continuous and relying on Gaussian kernels, and it has the particularity of enabling the representation of stimuli sampled from differently semantically structured symbolic spaces while maintaining the same representation shape (later referred as the *shape invariance property*), as opposed to the one-/multi-hot encoded (OHE/MHE) vector representation. While the SCS representation is inspired

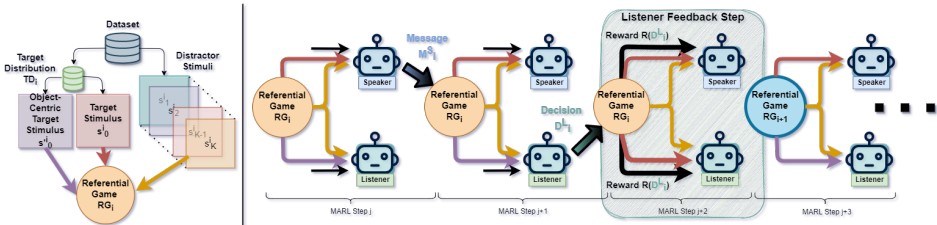

Figure 2: Left: Sampling of the necessary components to create the i-th RG ($RG_i$) of a meta-RG. The target stimulus (red) and the object-centric target stimulus (purple) are both sampled from the Target Distribution $TD_i$, a set of $O$ different stimuli representing the same latent semantic meaning. The latter set and a set of $K$ distractor stimuli (orange) are both sampled from a dataset of SCS-represented stimuli (**Dataset**), which is instantiated from the current episode's symbolic space, whose semantic structure is sampled out of the meta-distribution of available semantic structure over $N_{dim}$-dimensioned symbolic spaces. Right: Illustration of the resulting meta-RG with a focus on the i-th RG $RG_i$. The speaker agent receives at each step the target stimulus $s_0^i$ and distractor stimuli $(s_k^i)_{k \in [1;K]}$, while the listener agent receives an object-centric version of the target stimulus $s'^i_0$ or a distractor stimulus (randomly sampled), and other distractor stimuli $(s_k^i)_{k \in [1;K]}$, with the exception of the **Listener Feedback step** where the listener agent receives feedback in the form of the exact target stimulus $s_0^i$. The Listener Feedback step takes place after the listener agent has provided a decision $D_i^L$ about whether the target meaning is observed or not and in which stimuli is it instantiated, guided by the vocabulary-permutated message $M_i^S$ from the speaker agent.

by vectors sampled from VAE's latent spaces, this representation is not learned and is not aimed to help the agent performing its task. It is solely meant to make it possible to define a distribution over infinitely many semantic/symbolic spaces, while instantiating a BP for the agent to resolve. Indeed, contrary to OHE/MHE representation, observation of one stimulus is not sufficient to derive the nature of the underlying semantic space that the current episode instantiates. Rather, it is only via a kernel density estimation on multiple samples (over multiple timesteps) that the semantic space's nature can be inferred, thus requiring the agent to segregated and (re)combine information that is distributed over multiple observations. In other words, the benchmark instantiates a domain-agnostic BP. We provide in Appendix D.1 some numerical evidence to the fact that the SCS representation differentiates itself from the OHE/MHE representation because it instantiates a BP. Deriving the SCS representation from an idealised VAE's latent encoding of stimuli of any domain makes it a domain-agnostic representation, which is an advantage compared to previous benchmark because domain-specific information can therefore not be leveraged to solve the benchmark.

In details, the semantic structure of an $N_{dim}$-dimensioned symbolic space is the tuple $(d(i))_{i \in [1;N_{dim}]}$ where $N_{dim}$ is the number of latent/factor dimensions, $d(i)$ is the **number of possible symbolic values** for each latent/factor dimension $i$. Stimuli in the SCS representation are vectors sampled from the continuous space $[-1, +1]^{N_{dim}}$. In comparison, stimuli in the OHE/MHE representation are vectors from the discrete space $\{0, 1\}^{d_{OHE}}$ where $d_{OHE} = \Sigma_{i=1}^{N_{dim}} d(i)$ depends on the $d(i)$'s. Note that SCS-represented stimuli have a shape that does not depend on the $d(i)$'s values, this is the *shape invariance property* of the SCS representation (see Figure 4(bottom) for an illustration).

In the SCS representation, the $d(i)$'s do not shape the stimuli but only the semantic structure, i.e. representation and semantics are disentangled from each other. The $d(i)$'s shape the semantic by enforcing, for each factor dimension $i$, a partitionaing of the $[-1, +1]$ range into $d(i)$ value sections. Each partition corresponds to one of the $d(i)$ symbolic values available on the $i$-th factor dimension. Having explained how to build the SCS representation sampling space, we now describe how to sample stimuli from it. It starts with instantiating a specific latent meaning/symbol, embodied by latent values $l(i)$ on each factor dimension $i$, such that $l(i) \in [1; d(i)]$. Then, the $i$-th entry of the stimulus is populated with a sample from a corresponding Gaussian distribution over the $l(i)$-th partition of the $[-1, +1]$ range. It is denoted as $g_{l(i)} \sim \mathcal{N}(\mu_{l(i)}, \sigma_{l(i)})$, where $\mu_{l(i)}$ is the mean of the Gaussian distribution, uniformly sampled to fall within the range of the $l(i)$-th partition, and $\sigma_{l(i)}$ is the standard deviation of the Gaussian distribution, uniformly sampled over the range $[\frac{2}{12d(i)}, \frac{2}{6d(i)}]$. $\mu_{l(i)}$ and $\sigma_{l(i)}$ are sampled in order to guarantee (i) that the scale of the Gaussian distribution is large enough, but (ii) not larger than the size of the partition section it should fit in. Figure 3 shows an

example of such instantiation of the different Gaussian distributions over each factor dimensions'
$[-1, +1]$ range.

## 3.2 META-REFERENTIAL GAMES

Thanks to the *shape invariance property* of the SCS representation, once a number of latent/factor dimension $N_{dim}$ is choosen, we can synthetically generate many different semantically structured symbolic spaces while maintaining a consistent stimulus shape. This is critical since agents must be able to deal with stimuli coming from differently semantically structured $N_{dim}$-dimensioned symbolic spaces. In other words that are more akin to the meta-learning field, we can define a distribution over many kind of tasks, where each task instantiates a different semantic structure to the symbolic space our agent should learn to adapt to. Figure 2 highlights the structure of an episode, and its reliance on differently semantically structured $N_{dim}$-dimensioned symbolic spaces. Agents aim to coordinate efficiently towards scoring a high accuracy during the ZSCTs at the end of each RL episode. Indeed, a meta-RG is composed of two phases: a supporting phase where supporting stimuli are presented, and a querying/ZSCT phase where ZSCT-purposed RGs are played. During the querying phase, the presented target stimuli are novel combinations of the component values of the target stimuli presented during the supporting phase. Algorithms 4 and 5 contrast how a common RG differ from a meta-RG (in Appendix A). We emphasise that the supporting phase of a meta-RG does not involve updating the parameters/weights of the learning agents, since this is a meta-learning framework of the few-shot learning kind (compare positions and dependencies of lines 21 in Alg. 5 and 6 in Alg. 4). During the supporting phase, each RG involves a different target stimulus until all the possible component values on each latent/factor dimensions have been shown for at least $S$ shots (cf. lines $3 - 7$ in Alg. 5). While it amounts to at least $S$ different target stimulus being shown, the number of supporting-phase RG played remains far smaller than the number of possible training-purposed stimuli in the current episode's symbolic space/dataset. Then, the querying phase sees all the testing-purposed stimuli being presented.Emphasising further, during one single RL episode, both supporting/training-purposed and ZSCT/querying-purposed RGs are played, without the agent's parameters changing in-between the two phases, since learning CLBs involve agents adapting in an online/few-shot learning setting.

The semantic structure of the symbolic space is randomly sampled at the beginning of each episode (cf. lines $2 - 3$ in Alg. 5) The reward function proposed to both agents is null at all steps except on the $N+1$-th step. It is $+1$ if the listener agent decided correctly, and, during the querying phase only, $-2$ if incorrect (cf. line 21 in Alg. 5).

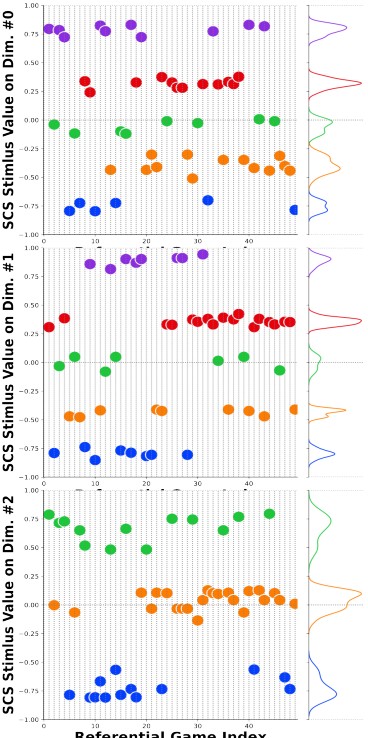

Figure 3: Visualisation of the SCS-represented stimuli (column) observed by the speaker agent at each RG over the course of one meta-RG, with $N_{dim} = 3$ and $d(0) = 5$, $d(1) = 5$, $d(2) = 3$. The supporting phase lasted for 19 RGs. For each factor dimension $i \in [0; 2]$, we present on the right side of each plot the kernel density estimations of the Gaussian kernels $\mathcal{N}(\mu_{l(i)}, \sigma_{l(i)})$ of each latent value available on that factor dimension $l(i) \in [1; d(i)]$. Colours of dots, used to represent the sampled value $g_{l(i)}$, imply the latent value $l(i)$'s Gaussian kernel from which said continuous value was sampled. As per construction, for each factor dimension, there is no overlap between the different latent values' Gaussian kernels.

**Vocabulary Permutation.** We bring the readers attention on the fact that simply changing the semantic structure of the symbolic space, is not sufficient to force MARL agents to adapt specifically to the instantiated symbolic space at each episode. Indeed, they can learn to cheat by relying on an episode-invariant (and therefore independent of the instantiated semantic structure) emergent language which would encode the continuous values of the SCS representation like an analog-to-digital converter would. This cheating language would consist of mapping a fine-enough partition of the $[-1, +1]$ range onto a fixed vocabulary in a bijective fashion (see Appendix C for more details). Therefore, in order to guard the MARL agents from making a cheating language emerge, we employ

a vocabulary permutation scheme (Cope and Schoots, 2021) that samples at the beginning of each episode/task a random permutation of the vocabulary symbols (cf. line 1 in Alg. 2).

**Richness of the Stimulus.** We further bridge the gap between Hill-RSC and Chaa-RSC by allowing the **number of object-centric samples** $O$ and the **number of shots** $S$ to be parameterized in the benchmark. $S$ represents the minimal number of times any given component value may be observed throughout the course of an episode. Intuitively, throughout their lifespan, an embodied observer may only observe a given component (e.g. the value 'blue', on the latent/factor dimension 'color') a limited number of times (e.g. one time within a 'blue car' stimulus, and another time within a 'blue cup' stimulus). These parameters allow the experimenters to account for both the Chaa-RSC's sampling density of the different stimulus components and Hill-RSC's diversity of viewpoints.

## 4 EXPERIMENTS

**Agent Architecture.** The architectures of the RL agents that we consider are detailed in Appendix B. Optimization is performed via an R2D2 algorithm(Kapturowski et al., 2018) and hyperparameters have been fine-tuned with the Sweep feature of Weights&Biases(Biewald, 2020). More details can be found in our open-source implementation[1]. Initial results show that the benchmark is too complex for our simple agents. Thus, following the work of Hill et al. (2020), we augment the agent with an auxiliary reconstruction task to help the agent in learning how to use its core memory module to some extent. The reconstruction loss consists of a mean squared-error between the stimuli observed by the agent at a given time step and the output of a prediction network which takes as input the current state of the core memory module after processing the current timestep stimuli.

### 4.1 LEARNING CLBs IS OUT-OF-REACH TO STATE-OF-THE-ART MARL

Playing a meta-RG, we highlight that, at each episode, the speaker aims to make emerge a new language (constructivity) and the listener aims to acquire it (receptivity) as fast as possible, before the querying-phase of the episode comes around. Critically, we assume that both agents must perform while abiding by the principles of CLBs as it is the only resolution approach. Indeed, there is no success without a generalizing and easy-to-learn emergent language, or, in other words, a (linguistically) compositional emergent language (Brighton and Kirby, 2001; Brighton, 2002). Thus, we investigate whether agents are able to coordinate to learn to perform CLBs from scratch, which is tantamount to learning receptivity and constructivity aspects of CLBs in parallel.

Table 1: Meta-RG ZSCT and Ease-of-Acquisition (EoA) ZSCT accuracies and linguistic compositionality measures ($\% \pm$ s.t.d.) for the multi-agent context after a sampling budget of $500k$. The last column shows linguist results when evaluating the Posdis-Speaker (PS).

| | Shots | | PS |
|---|---|---|---|
| Metric | $S = 1$ | $S = 2$ | |
| $Acc_{\text{ZSCT}} \uparrow$ | $53.56 \pm 4.68$ | $51.58 \pm 2.24$ | N/A |
| $Acc_{\text{EoA}} \uparrow$ | $50.56 \pm 8.75$ | $50.63 \pm 5.78$ | N/A |
| topsim $\uparrow$ | $29.58 \pm 16.76$ | $21.32 \pm 16.55$ | $96.67 \pm 0$ |
| posdis $\uparrow$ | $23.74 \pm 20.75$ | $13.83 \pm 12.75$ | $92.03 \pm 0$ |
| bosdis $\uparrow$ | $25.61 \pm 22.88$ | $19.14 \pm 17.26$ | $11.57 \pm 0$ |

**Evaluation.** We report on 3 random seeds of an LSTM-based model augmented with both the *Value Decomposition Network* (Sunehag et al., 2017) and the *Simplified Action Decoder* approach (Hu and Foerster, 2019), in the task with $N_{dim} = 3, V_{min} = 2, V_{max} = 5, O = 4$, and $S = 1$ or $2$. As we assume no success without emergence of a (linguistically) compositional language, we measure the linguistic compositionality profile of the emerging languages by, firstly, freezing the speaker agent's internal state (i.e. LSTM's hidden and cell states) at the end of an episode and query what would be its subsequent utterances for all stimuli in the latest episode's dataset (see Figure 2), and then compute the different compositionality metrics on this collection of utterances. We compare the compositionality profile of the emergent languages to that of a compositional language, in the sense of the **posdis** compositionality metric (Chaabouni et al., 2020) (see Figure 4(left) and Table 3 in Appendix B.2). This language is produced by a fixed, rule-based agent that we will refer to as the Posdis-Speaker (PS). Similarly, after the latest episode ends and the speaker agent's internal state is frozen, we evaluate the EoA of the emerging languages by training a **new, non-meta/common**

---

[1]HIDDEN-FOR-REVIEW-PURPOSE

**listener agent** for 512 epochs on the latest episode's dataset with the frozen speaker agent using a *descriptive-only/object-centric* **common** RG and report its ZSCT accuracy (see Algorithm 3).

**Results.** Table 1 shows the performance and compositionality of the behaviours in the multi-agent context. As $Acc_{\text{ZSCT}}$ is around chance-level ($50\%$), the meta-RL agents fail to coordinate together, despite the simplicity of the setting. This shows that learning CLBs from scratch is currently out-of-reach to state-of-the-art MARL agents, and therefore show the importance of our benchmark. As the linguistic compositionality measures are very low compared to the PS agent, and since the chance-leveled $Acc_{\text{EoA}}$ implies that the emerging languages are not easy to learn, it leads us to think that the poor multi-agent performance is due to the lack of compositional language emergence.

## 4.2 RECEPTIVITY ASPECTS OF CLBS CAN BE LEARNED SUB-OPTIMALLY

Seeing that the multi-agent benchmark is out of reach to state-of-the-art cooperative MARL agents, we investigate a simplification along two axises. Firstly, we simplify to a single-agent RL problem by instantiating a fixed, rule-based agent as the speaker, which should remove any issues related to agents learning in parallel to coordinate. Secondly, we use the Posdis-Speaker agent, which should remove any issues related to the emergence of assumed-necessary compositional languages, which corresponds to the constructivity aspects of CLBs. These simplifications allow us to focus our investigation on the receptivity aspects of CLBs, which relates to the ability from the listener agent to acquire and leverage a newly-encountered compositional language at each episode.

**Hypotheses.** We show in Appendix D.1 that the SCS representation instantiates a BP even when $O = 1$. Thus, we suppose that when $O$ increases the BP's complexity increases, as the more object-centric samples there are, the greater the number of steps of relational responding are involved to estimate the size of each Gaussian distribution. Thus, it would stand to

Table 2: ZSCT accuracies (%) for the LSTM-based agent.

| | Shots | | |
|---|---|---|---|
| Samples | $S = 1$ | $S = 2$ | $S = 4$ |
| $O = 1$ | $62.23 \pm 3.67$ | $73.45 \pm 2.39$ | $74.94 \pm 2.25$ |
| $O = 4$ | $62.80 \pm 0.75$ | $62.55 \pm 1.66$ | $60.20 \pm 2.19$ |
| $O = 16$ | $64.92 \pm 1.70$ | $61.96 \pm 2.00$ | $61.83 \pm 2.11$ |

reason to expect the ZSCT accuracy to decrease when $O$ increases (Hyp. 1). On the other hand, we would expect that increasing $S$ would provide the learning agent with a denser sampling (in order to fulfill Chaa-RSC (ii)) , and thus systematicity is expected to increase as $S$ increases (Hyp. 2). Indeed, increasing $S$ is tantamount to increasing the number of opportunities given to the agents to estimate the size of each Gaussian distribution, thus relaxing the instantiated BP's complexity.

**Evaluation.** We report ZSCT accuracies on LSTM-based models (6 random seeds per settings) with $N_{dim} = 3$ and $V_{min} = 2, V_{max} = 5$.

**Results.** Table 2 shows the ZSCT accuracies while varying the number of object-centric samples $O$, and the number of shots $S$. The chance threshold is $50\%$. When $S = 1$, increasing $O$ is surprisingly correlated with non-significant increases in systematicity. On the otherhand, when $S > 1$, accuracy distributions stay similar or decrease while $O$ increases. Thus, overall, Hyp. 1 tends to be validated. Regarding Hyp. 2, when $O = 1$, increasing $S$ (and with it the density of the sampling of the input space, i.e. Chaa-RSC (ii)) correlates with increases in systematicity. Thus, despite the difference of settings between common RG, in Chaabouni et al. (2020), and meta-RG here, we retrieve a similar result that Chaa-RSC promotes systematicity. On the other hand, our results show a statistically significant distinction between BPs of complexity associated with $O > 1$ and those associated with $O = 1$. Indeed, when $O > 1$, our results contradict Hyp.2 since accuracy distributions remain the same or decrease when $S$ increases. We explain these results based on the fact that increasing $S$ also increases the length of each RL episode, thus the 'algorithm' learned by LSTM-based agents fails to adequately estimate Gaussian kernel densities associated with each component value, possibly because of LSTMs' notorious difficulty with integrating/binding information from past to present inputs over long dependencies. We investigate this issue further in Appendix D.2.1.

## 5 DISCUSSION & RELATED WORKS

We provided experimental evidence that learning CLBs in an MARL setting is out-of-reach of state-of-the-art approaches, and that the bottleneck may be located in the constructivity aspects of CLBs

(embodied by the ability from the speaker agent to make compositional languages emerge at each episode) since artificial agents are able to perform sub-optimally when it comes to receptivity aspects of CLBs (embodied by the ability from the listener agent to acquire and leverage a compositional language at each episode). These results validate the need for our benchmark and they highlight that our main efforts should be focused on constructivity aspects of CLBs.

Our experiments took place in simplest instances of our benchmark, and made use of MARL approaches not specifically designed for the learning of CLBs. Subsequent works ought to involve novel, specifically-designed approaches towards addressing more complex instances of our benchmark.

**Compositional Behaviours vs CLBs.** The learning of compositional behaviours (CBs) is one of the central study in language grounding with benchmarks like SCAN (Lake and Baroni, 2018) and gSCAN (Ruis et al., 2020), as well as in the subfield of Emergent Communication (see Brandizzi (2023) for complete review), but none investigates nor allow testing for CLBs. Thus, towards developing artificial agents with greater human-cooperation abilities, our benchmark aims to fill in this gap. Outside of the subfield of Emergent Communication, CLBs have actually been used as a training paradigm towards enabling systematic compositional behaviours in the works of Lake (2019) and Lake and Baroni (2023). They propose a meta-learning extension to the sequence-to-sequence learning setting towards enabling systematicity, in the sense of CBs, on par with human performance. On the contrary to our work, abilities towards CBs are evaluated after using CLBs as a training tool only. Given the demonstrated impact of CLBs, we propose a CLB-principled benchmark for evaluation of CLBs abilities themselves in order to address research questions related to this novel research direction.

**Symbolic Behaviours.** Following Santoro et al. (2021)'s definition of symbolic behaviours, there is currently no specifically-principled benchmark to evaluate systematically whether artificial agents are able to engage into any symbolic behaviours. Thus, our benchmark aims to fill this gap.

**Binding Problem.** Similarly, following the definition and argumentation of Greff et al. (2020), while most challenging benchmark, especially in computer vision, instantiates a version of the BP, there is currently, to our knowledge, no principled benchmarks that specifically address the question whether the BP can be solved by artificial agents. Thus, not only does our benchmark fill that gap, but it also instantiate a domain-agnostic version of the BP, which is critical in order to ascertain the external validity of conclusions that may be drawn from it. Indeed, making the benchmark domain-agnostic guards us against any confounder that could make the task solvable without solving the BP, e.g. by gaming some possibly-domain-specific aspect (Chollet, 2019). We propose experimental evidence in Appendix D.1 that our benchmark does instantiate a BP.

## 6  CONCLUSION

In order to build artificial agents able to collaborate with human beings, we have proposed a novel benchmark to investigate artificial agents abilities at learning CLBs. Our proposed benchmark casts the problem of learning CLBs as a meta-reinforcement learning problem, using our proposed extension to RGs, entitled Meta-Referential Games, containing at its core an instantiation of the BP. It is made possible by a novel representation scheme, entitled the Symbolic Continuous Stimulus (SCS) representation, which is inspired from the vectors sampled in the latent space of VAE-like unsupervised learning approaches.

Previous literature identified richness of the stimuli as an important condition for agents to exhibit systematicity like the one required when it comes to CLBs. We provide a synthesis of the different RSCs that has been previously formulated, and made the parameters to this synthesis definition available in our proposed benchmark.

Finally, we provided baseline results for both the multi-agent tasks and the single-agent listener-focused tasks of learning CLBs in the context of our proposed benchmark. We provided an analysis of the behaviours in the multi-agent context highlighting the complexity for the speaker agent to invent a linguistically compositional language. When the language is already compositional to some extent, then we found that the listener is able to acquire it and coordinate with the speaker towards performing sub-optimally, in some of the simplest settings possible. Thus, overall, our results show that our proposed benchmark is currently out of reach for those agents. Thus, we hope it will spur the research community towards developing more capable artificial agents.

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

# A  ON ALGORITHMIC DETAILS OF META-REFERENTIAL GAMES

In this section, we detail algorithmically how Meta-Referential Games differ from common RGs. We start by presenting in Algorithm 4 an overview of the common RGs, taking place inside a common supervised learning loop, and we highlight the following:

1. (i) preparation of the data on which the referential game is played (highlighted in green),

2. (ii) elements pertaining to playing a RG (highlighted in blue),

3. (iii) elements pertaining to the **supervised learning loop** (highlighted in purple).

Helper functions are detailed in Algorithm 1, 2 and 3. Next, we can now show in greater and contrastive details the Meta-Referential Game algorithm in Algorithm 5, where we highlight the following:

1. (i) preparation of the data on which the referential game is played (highlighted in green),

2. (ii) elements pertaining to playing a RG (highlighted in blue),

3. (iii) elements pertaining to the **meta-learning loop** (highlighted in purple).

4. (iv) elements pertaining to setup of a Meta-Referential Game (highlighted in red).

---

**Algorithm 1:** Helper function : DataPrep

**Given**   :
  - a target stimuli $s_0$,
  - a dataset of stimuli Dataset,
  - $O$ : Number of Object-Centric samples in each Target Distribution over stimuli $TD(\cdot)$.
  - $K$ : Number of distractor stimuli to provide to the listener agent.
  - FullObs : Boolean defining whether the speaker agent has full (or partial) observation.
  - DescrRatio : Descriptive ratio in the range $[0, 1]$ defining how often the listener agent is observing the same semantic as the speaker agent.

1  $s_0', D^{Target} \leftarrow s_0, 0$;
2  **if** $random(0, 1) > DescrRatio$ **then**
3  $\quad$ $s_0' \sim \text{Dataset} - TD(s_0)$;; $\quad$ /* Exclude target stimulus from listener's observation ...   */
4  $\quad$ $D^{Target} \leftarrow K + 1$;; $\quad$ /* ...   and expect it to decide accordingly.   */
5  **end**
6  **else if** $O > 1$ **then**
7  $\quad$ Sample an Object-Centric distractor $s_0' \sim TD(s_0)$;
8  **end**
9  Sample $K$ distractor stimuli from $\text{Dataset} - TD(s_0)$: $(s_i)_{i \in [1,K]} \sim \text{Dataset} - TD(s_0)$;
10  $Obs_{\text{Speaker}} \leftarrow \{s_0\}$; **if** *FullObs* **then**
11  $\quad$ $Obs_{\text{Speaker}} \leftarrow \{s_0\} \cup \{s_i | \forall i \in [1, K]\}$;
12  **end**
13  $Obs_{\text{Listener}} \leftarrow \{s_0'\} \cup \{s_i | \forall i \in [1, K]\}$;
/* Shuffle listener observations and update index of target decision: $\quad\quad\quad\quad$ */
14  $Obs_{\text{Listener}}, D^{Target} \leftarrow Shuffle(Obs_{\text{Listener}}, D^{Target})$;
**Output**  : $Obs_{\text{Speaker}}, Obs_{\text{Listener}}, D^{Target}$;

---

**Algorithm 2:** Helper function : MetaRGDatasetPreparation

**Given :**
- $V$ : Vocabulary (finite set of tokens available),
- $N_{\mathrm{dim}}$ : Number of attribute/factor dimensions in the symbolic spaces,
- $V_{min}$ : Minimum number of possible values on each attribute/factor dimensions in the symbolic spaces,
- $V_{max}$ : Maximum number of possible values on each attribute/factor dimensions in the symbolic spaces,

1 Initialise random permutation of vocabulary: $V' \leftarrow RandomPerm(V)$
2 Sample semantic structure: $(d(i))_{i \in [1, N_{\mathrm{dim}}]} \sim \mathcal{U}(V_{min}; V_{max})^{N_{\mathrm{dim}}}$;
3 Generate symbolic space/dataset $D\big((d(i))_{i \in [1, N_{\mathrm{dim}}]}\big)$;
4 Split dataset into supporting set $D^{\mathrm{support}}$ and querying set $D^{\mathrm{query}}$ $(((d(i))_{i \in [1, N_{\mathrm{dim}}]})$ is omitted for readability);

**Output :** $V', D\big((d(i))_{i \in [1, N_{\mathrm{dim}}]}\big), D^{\mathrm{support}}, D^{\mathrm{query}}$;

---

**Algorithm 3:** Helper function : PlayRG

**Given :**
- Speaker and Listener agents,
- Set of speaker observations $Obs_{\mathrm{Speaker}}$,
- Set of listener observations $Obs_{\mathrm{Listener}}$,
- $N$ : Number of communication rounds to play,
- $L$ : Maximum length of each message,
- $V$ : Vocabulary (finite set of tokens available),

1 Compute message $M^S = \mathrm{Speaker}(Obs_{\mathrm{Speaker}}|\emptyset)$;
2 Initialise Communication Channel History: $\mathrm{CommH} \leftarrow [M^S]$;
3 **for** *round = 0, N* **do**
4      Compute Listener's reply $M^L_{\mathrm{round}}, \_ = \mathrm{Listener}(Obs_{\mathrm{Listener}}|\mathrm{CommH})$;
5      $\mathrm{CommH} \leftarrow \mathrm{CommH} + [M^L_{\mathrm{round}}]$;
6      Compute Speaker's reply $M^S_{\mathrm{round}} = \mathrm{Speaker}(Obs_{\mathrm{Speaker}}|\mathrm{CommH})$;
7      $\mathrm{CommH} \leftarrow \mathrm{CommH} + [M^S_{\mathrm{round}}]$;
8 **end**
9 Compute listener decision $\_, D^L = \mathrm{Listener}(Obs_{\mathrm{Listener}}|\mathrm{CommH})$;

**Output :** Listener's decision $D^L$, Communication Channel History CommH;

---

**Algorithm 4:** Common Referential Game inside a Common Supervised Learning Loop

**Given** :
- a dataset of stimuli $Dataset$,
- a set of hyperparameters defining the RG:
  - $O$ : Number of Object-Centric samples in each Target Distribution over stimuli $TD(\cdot)$.
  - $N$ : Number of communication rounds to play.
  - $L$ : Maximum length of each message.
  - $V$ : Vocabulary (finite set of tokens available).
  - $K$ : Number of distractor stimuli to provide to the listener agent.
  - FullObs : Boolean defining whether the speaker agent has full (or partial) observation.
  - DescrRatio : Descriptive ratio in the range $[0, 1]$ defining how often the listener agent is observing the same semantic as the speaker agent.
  - $\mathcal{L}$ : Loss function to use in the agents update.

**Initialize** :
- Speaker$(\cdot)$ and Listener$(\cdot)$ agents.

1 Systematically split $Dataset$ into training and testing dataset, $D^{\text{train}}$ and $D^{\text{test}}$;
2 **for** $epoch = 1, N_{epoch}$ **do**
3    **for** *target stimulus* $s_0 \in D^{train}$ **do**
      `/* Preparation of observations and target decision: */`
4       $Obs_{\text{Speaker}}, Obs_{\text{Listener}}, D^{Target} \leftarrow DataPrep(\text{Dataset}, s_0, O, K, \text{FullObs}, \text{DescrRatio})$
      `/* Play Referential Game: */`
5       $D^L, \_ = \text{PlayRG}(\text{Speaker}, \text{Listener}, Obs_{\text{Speaker}}, Obs_{\text{Listener}}, N, L, V)$;
      `/* Supervised Learning Parameters Update on Training Stimulus Only: */`
6       Update both speaker and listener agents' parameters using the loss $\mathcal{L}(D^{Target}, D^L)$;
7    **end**
8    Initialise ZSCT accuracy: $Acc_{\text{ZSCT}} \leftarrow 0$;
9    **for** *target stimulus* $s_0 \in D^{test}$ **do**
      `/* Preparation of observations and target decision: */`
10       $Obs_{\text{Speaker}}, Obs_{\text{Listener}}, D^{Target} \leftarrow DataPrep(\text{Dataset}, s_0, O, K, \text{FullObs}, \text{DescrRatio})$
      `/* Play Referential Game: */`
11       $D^L, \_ = \text{PlayRG}(\text{Speaker}, \text{Listener}, Obs_{\text{Speaker}}, Obs_{\text{Listener}}, N, L, V)$;
      `/* Update ZSCT Accuracy: */`
12       $Acc_{\text{ZSCT}} \leftarrow \text{Update}(Acc_{\text{ZSCT}}, D^{Target}, D^L)$;
13    **end**
14 **end**

---

---

**Algorithm 5:** Meta-Referential Game inside a Meta-Learning Loop

**Given** :
- $N_{episode}$, $N_{\dim}$ : Number of episodes, and number of attribute/factor dimensions,
- $S$ : Minimum number of Shots over which each possible value on each attribute/factor dimension ought to be observed by the agents (as part of a target stimulus).
- $V_{min}, V_{max}$ : Minimum and maximum number of possible values on each attribute/factor dimensions in the symbolic spaces,
- $TSS(\mathcal{D}, \mathcal{S}, S)$ : Target stimulus sampling function which samples from dataset $\mathcal{D}$, given a set of previously sampled stimuli $\mathcal{S}$, while maximising the likelihood that each possible value on each attribute/factor dimension are sampled at least $S$ times.
- a set of hyperparameters defining the RG:
    - $O$ : Number of Object-Centric samples in each Target Distribution over stimuli $TD(\cdot)$.
    - $N$ : Number of communication rounds to play.
    - $L$ : Maximum length of each message.
    - $V$ : Vocabulary (finite set of tokens available).
    - $K$ : Number of distractor stimuli to provide to the listener agent.
    - FullObs : Boolean defining whether the speaker agent has full (or partial) observation.
    - DescrRatio : Descriptive ratio in the range $[0, 1]$ defining how often the listener agent is observing the same semantic as the speaker agent.

**Initialize :**
- Speaker$(\cdot)$ and Listener$(\cdot)$ agents.

1 **for** $episode = 1, N_{episode}$ **do**
    /* **Preparation of the symbolic space/dataset:** */
2     $V', D_{\text{episode}}, D_{\text{episode}}^{\text{support}}, D_{\text{episode}}^{\text{query}} \leftarrow MetaRGDatasetPreparation(V, N_{\dim}, V_{\min}, V_{\max})$;
3     Initialise set of sampled supporting stimuli: $\mathcal{S}^{\text{support}} \leftarrow \emptyset$;
4     **repeat**
5         Sample training-purposed target stimulus $s_0^i \sim TSS(D_{\text{episode}}^{\text{support}}, \mathcal{S}^{\text{support}}, S)$
6         $\mathcal{S}^{\text{support}} \leftarrow \mathcal{S}^{\text{support}} \cup \{s_0^i\}; i \leftarrow i + 1$;
7     **until** *all values on each attribute/factor dimension have been instantiated at least $S$ times*;
8     Initialise RG index: $i \leftarrow 0$;
    /* **Supporting Phase:** */
9     **for** *target stimulus $s_0^i \in \mathcal{S}^{support}$* **do**
10         $Obs_{\text{Speaker}}^i, Obs_{\text{Listener}}^i, D_i^{Target} \leftarrow DataPrep(D_{\text{episode}}^{\text{support}}, s_0^i, O, K, \text{FullObs}, \text{DescrRatio})$;
11         $D_i^L, CommH_i = \text{PlayRG}(\text{Speaker}, \text{Listener}, Obs_{\text{Speaker}}^i, Obs_{\text{Listener}}^i, N, L, V')$;
12         $\_, \_ = \text{Listener}(Obs_{Speaker}^i | CommH_i)$ ;     /* **Listener-Feedback Step** */
13     **end**
    /* **Querying/ZSCT Phase:** */
14     Initialise ZSCT accuracy: $Acc_{\text{ZSCT}} \leftarrow 0$;
15     **for** *target stimulus $s_0^i \in D_{episode}^{query}$* **do**
16         $Obs_{\text{Speaker}}^i, Obs_{\text{Listener}}^i, D_i^{Target} \leftarrow DataPrep(D_{\text{episode}}, s_0^i, O, K, \text{FullObs}, \text{DescrRatio})$;
17         $D_i^L, CommH_i = \text{PlayRG}(\text{Speaker}, \text{Listener}, Obs_{\text{Speaker}}^i, Obs_{\text{Listener}}^i, N, L, V')$;
18         $\_, \_ = \text{Listener}(Obs_{Speaker}^i | CommH_i)$ ;     /* **Listener-Feedback Step** */
        /* **Update ZSCT Accuracy:** */
19         $Acc_{\text{ZSCT}} \leftarrow \text{Update}(Acc_{\text{ZSCT}}, D_i^{Target}, D_i^L); \; i \leftarrow i + 1$;
20     **end**
    /* **Meta-Learning Parameters Update on Whole Episode:** */
21     Update both agents using rewards $R_i = \begin{cases} 1 & \text{if } D_i^{Target} == D_i^L \\ 0 & \text{otherwise, during supporting phase} \\ -2 & \text{otherwise, during querying phase} \end{cases}$
22 **end**

---

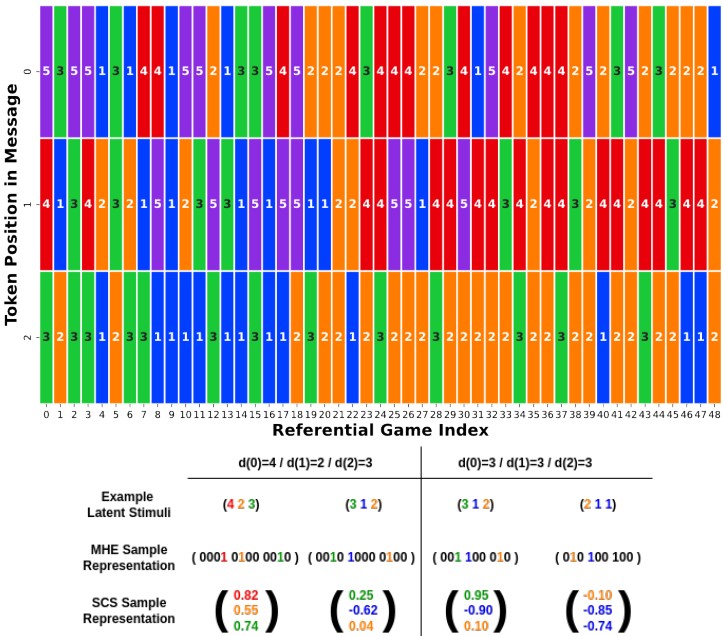

Figure 4: **Top:** visualisation on each column of the messages sent by the posdis-compositional rule-based speaker agent over the course of the episode presented in Figure 3. Colours are encoding the information of the token index, as a visual cue. **Bottom:** OHE/MHE and SCS representations of example latent stimuli for two differently-structured symbolic spaces with $N_{dim} = 3$, i.e. on the left for $d(0) = 4$, $d(1) = 2$, $d(2) = 3$, and on the right for $d(0) = 3$, $d(1) = 3$, $d(2) = 3$. Note the shape invariance property of the SCS representation, as its shape remains unchanged by the change in semantic structure of the symbolic space, on the contrary to the OHE/MHE representations.

## B    AGENT ARCHITECTURE & TRAINING

The baseline RL agents that we consider use a 3-layer fully-connected network with 512, 256, and finally 128 hidden units, with ReLU activations, with the stimulus being fed as input. The output is then concatenated with the message coming from the other agent in a OHE/MHE representation, mainly, as well as all other information necessary for the agent to identify the current step, i.e. the previous reward value (either $+1$ and $0$ during the training phase or $+1$ and $-2$ during testing phase), its previous action in one-hot encoding, an OHE/MHE-represented index of the communication round (out of $N$ possible values), an OHE/MHE-represented index of the agent's role (speaker or listener) in the current game, an OHE/MHE-represented index of the current phase (either 'training' or 'testing'), an OHE/MHE representation of the previous RG's result (either success or failure), the previous RG's reward, and an OHE/MHE mask over the action space, clarifying which actions are available to the agent in the current step. The resulting concatenated vector is processed by another 3-layer fully-connected network with 512, 256, and 256 hidden units, and ReLU activations, and then fed to the core memory module, which is here a 2-layers LSTM (Hochreiter and Schmidhuber, 1997) with 256 and 128 hidden units, which feeds into the advantage and value heads of a 1-layer dueling network (Wang et al., 2016).

Table 5 highlights the hyperparameters used for the learning agent architecture and the learning algorithm, R2D2(Kapturowski et al., 2018). More details can be found, for reproducibility purposes, in our open-source implementation at HIDDEN-FOR-REVIEW-PURPOSE.

Training was performed for each run on 1 NVIDIA GTX1080 Ti, and the average amount of training time for a run is 18 hours for LSTM-based models, 40 hours for ESBN-based models, and 52 hours for DCEM-based models.

## B.1 ESBN & DCEM

The ESBN-based and DCEM-based models that we consider have the same architectures and parameters than in their respective original work from Webb et al. (2020) and Hill et al. (2020), with the exception of the stimuli encoding networks, which are similar to the LSTM-based model.

## B.2 RULE-BASED SPEAKER AGENT

The rule-based speaker agents used in the single-agent task, where only the listener agent is a learning agent, speaks a compositional language in the sense of the posdis metric (Chaabouni et al., 2020), as presented in Table 3 for $N_{dim} = 3$, a maximum sentence length of $L = 4$, and vocabulary size $|V| >= max_i d(i) = 5$, assuming a semantical space such that $\forall i \in [1, 3], d(i) = 5$.

## C CHEATING LANGUAGE

The agents can develop a cheating language, cheating in the sense that it could be episode/task-invariant (and thus semantic structure invariant). This emerging cheating language would encode the continuous values of the SCS representation like an analog-to-digital converter would, by mapping a fine-enough partition of the $[-1, +1]$ range onto the vocabulary in a bijective fashion.

For instance, for a vocabulary size $\|V\| = 10$, each symbol can be unequivocally mapped onto $\frac{2}{10}$-th increments over $[-1, +1]$, and, by communicating $N_{dim}$ symbols (assuming $N_{dim} \leq L$), the speaker agents can communicate to the listener the (digitized) continuous value on each dimension $i$ of the SCS-represented stimulus. If $max_j d(j) \leq \|V\|$ then the cheating language is expressive-enough for the speaker agent to digitize

Table 3: Examples of the latent stimulus to language utterance mapping of the posdis-compositional rule-based speaker agent. Note that token 0 is the EoS token.

| Latent Dims | | | Comp. Language |
|---|---|---|---|
| #1 | #2 | #3 | Tokens |
| 0 | 1 | 2 | 1, 2, 3, 0 |
| 1 | 3 | 4 | 2, 4, 5, 0 |
| 2 | 5 | 0 | 3, 6, 1, 0 |
| 3 | 1 | 2 | 4, 2, 3, 0 |
| 4 | 3 | 4 | 5, 4, 5, 0 |

all possible stimulus without solving the binding problem, i.e. without inferring the semantic structure. Similarly, it is expressive-enough for the listener agent to convert the spoken utterances to continuous/analog-like values over the $[-1, +1]$ range, thus enabling the listener agent to skirt the binding problem when trying to discriminate the target stimulus from the different stimuli it observes.

## D FURTHER EXPERIMENTS:

### D.1 ON THE BP INSTANTIATED BY THE SCS REPRESENTATION

**Hypothesis.** The SCS representation differs from the OHE/MHE one primarily in terms of the binding problem (Greff et al., 2020) that the former instantiates while the latter does not. Indeed, the semantic structure can only be inferred after observing multiple SCS-represented stimuli. We hypothesised that it is via the *dynamic binding of information* extracted from each observations that an estimation of a density distribution over each dimension $i$'s $[-1, +1]$ range can be performed. And, estimating such density distribution is tantamount to estimating the number of likely gaussian distributions that partition each $[-1, +1]$ range.

**Evaluation.** Towards highlighting that there is a binding problem taking place, we show results of baseline RL

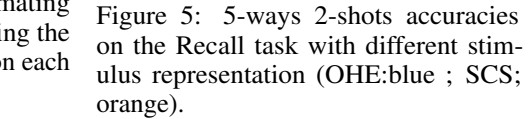

Figure 5: 5-ways 2-shots accuracies on the Recall task with different stimulus representation (OHE:blue ; SCS; orange).

agents (similar to main experiments in Section 4) evaluated on a simple single-agent recall task. The Recall task structure borrows from few-shot learning tasks as it presents over 2 shots all the stimuli of the instantiated symbolic space (not to be confused with the case for Meta-RG where all the latent/factor dimensions' values are being presented over $S$ shots – Meta-RGs do not necessarily

sample the whole instantiated symbolic space at each episode, but the Recall task does). Each shot consists of a series of recall games, one for each stimulus that can be sampled from an $N_{dim} = 3$-dimensioned symbolic space. The semantic structure $(d(i))_{i \in [1;N_{dim}]}$ of the symbolic space is randomly sampled at the beginning of each episode, i.e. $d(i) \sim \mathcal{U}(2;5)$, where $\mathcal{U}(2;5)$ is the uniform discrete distribution over the integers in $[2;5]$, and the number of object-centric samples is $O = 1$, in order to remove any confounder from object-centrism.

Each recall game consists of two steps: in the first step, a stimulus is presented to the RL agent, and only a *no-operation* (NO-OP) action is made available, while, on the second step, the agent is asked to infer/recall the **discrete** $l(i)$ **latent value** (as opposed to the representation of it that it observed, either in the SCS or OHE/MHE form) that the previously-presented stimulus had instantiated, on a given $i$-th dimension, where value $i$ for the current game is uniformly sampled from $\mathcal{U}(1; N_{dim})$ at the beginning of each game. The value of $i$ is communicated to the agent via the observation on this second step of different stimulus that in the first step: it is a zeroed out stimulus with the exception of a 1 on the $i$-th dimension on which the inference/recall must be performed when using SCS representation, or over all the OHE/MHE dimensions that can encode a value for the $i$-th latent factor/attribute when using the OHE/MHE representation. On the second step, the agent's available action space now consists of discrete actions over the range $[1; max_j d(j)]$, where $max_j d(j)$ is a hyperparameter of the task representing the maximum number of latent values for any latent/factor dimension. In our experiments, $max_j d(j) = 5$. While the agent is rewarded at each game for recalling correctly, we only focus on the performance over the games of the second shot, i.e. on the games where the agent has theoretically received enough information to infer the density distribution over each dimension $i$'s $[-1, +1]$ range. Indeed, observing the whole symbolic space once (on the first shot) is sufficient (albeit not necessary, specifically in the case of the OHE/MHE representation).

**Results.** Figure 3(right) details the recall accuracy over all the games of the second shot of our baseline RL agent throughout learning. There is a large gap of asymptotic performance depending on whether the Recall task is evaluated using OHE/MHE or SCS representations. We attribute the poor performance in the SCS context to the instantiation of a BP. We note again that during those experiments the number of object-centric samples was kept at $O = 1$, thus emphasising that the BP is solely depending on the use of the SCS representation and does not require object-centrism.

## D.2 ON THE IDEALLY-DISENTANGLED-NESS OF THE SCS REPRESENTATION

In this section, we verify our hypothesis that the SCS representation yields ideally-disentangled stimuli. We report on the **FactorVAE Score** Kim and Mnih (2018), the Mutual Information Gap (**MIG**) Chen et al., and the **Modularity Score** Ridgeway and Mozer (2018) as they have been shown to be part of the metrics that correlate the least among each other (Locatello et al., 2020), thus representing different desiderata/definitions for disentanglement. We report on the $N_{dim} = 3$-dimensioned symbolic spaces with $\forall j, d(j) = 5$. The measurements are of $100.0\%$, $94.8$, and $98.9\%$ for, respectively, the FactorVAE Score, the MIG, and the Modularity Score, thus validating our design hypothesis about the SCS representation. We remark that, while it is not the case of the FactorVAE Score (possibly due to its reliance on a deterministic classifier), the MIG and Modularity Score are sensitive to the number of object-centric samples, decreasing as low as $64.4\%$ and $66.6\%$ for $O = 1$.

### D.2.1 IMPACTS OF MEMORY AND SAMPLING BUDGET

**Hypothesis.** Seeing the discrepancies around the impact of $S$ and $O$, and given both (i) the fact that increasing $S$ inevitably increases the length of the episode (thus making it difficult for LSTMs as they notoriously fail to inte-

Table 4: **Bottom:** ZSCT accuracies (%) for $S = 1$ and $O = 1$, with a $10M$ observation budget.

|  | LSTM | ESBN | DCEM |
|---|---|---|---|
| Acc.(%) | $86.00 \pm 0.14$ | $89.35 \pm 2.77$ | $81.90 \pm 0.59$ |

grate information from past to present inputs over long dependencies) and (ii) our results goading us to think that increasing $S$ when $O > 1$ is confusing LSTM-based agent rather than helping them, we hypothesise that using agents with more efficient memory-addressing mechanism (e.g. attention) and greater algorithm-learning abilities (e.g. with explicit memory) would help the agent get past the identified hurdles. While we leave it to subsequent work to investigate the impact of more sophisticated core memory when $O > 1$, we here propose off-the-shelf baseline results in the simplest setting and with a greater sampling budget. Indeed, we hypothesis that **a denser sampling**

**of the space of all symbolic spaces** ought to be beneficial, as we extrapolate Chaa-RSCH (ii) to its meta-learning version, and avoid the issues pertaining to long dependencies.

**Evaluation.** We report on architectures based on the Emergent Symbol Binding Network (ESBN) (Webb et al., 2020) and the Dual-Coding Episodic Memory (DCEM) (Hill et al., 2020) (resp. 3 and 2 random seeds), in the setting of $N_{dim} = 3$, $V_{min} = 2$ and $V_{max} = 3$, with a sampling budget of $10M$ observations, as opposed to $5M$ previously.

**Results.** Table 4 presents the results, which are slightly in favour of the ESBN-based model. As it is built over the LSTM-based one, the memory scheme may be bypassed until it provides advantages with a simple-enough mechanism, as opposed to the case of the DCEM-based model.

## D.3 IMPACT OF DIFFERENT CORE MEMORY MODULES

In the following, we investigate and compare the performance when using an LSTM (Hochreiter and Schmidhuber, 1997) or a Differentiable Neural Computer (DNC) (Graves et al., 2016) as core memory module.

Regarding the auxiliary reconstruction loss, in the case of the LSTM, the hidden states are used, while in the case of the DNC, the memory is used as input to the prediction network. Figure 6b shows the stimulus reconstruction accuracies for both architectures, highlighting a greater data-efficiency (and resulting asymptotic performance in the current observation budget) of the LSTM-based architecture, compared to the DNC-based one.

Figure 6a shows the 4-ways (3 distractors descriptive RG variant) ZSCT accuracies of the different agents throughout learning. The ZSCT accuracy is the accuracy over testing-purpose stimuli only, after the agent has observed for two consecutive times (i.e. $S = 2$) the supportive training-purpose stimuli for the current episode. The DNC-based architecture has difficulty learning how to use its memory, even with the use of the auxiliary reconstruction loss, and therefore it utterly fails to reach better-than-chance ZSCT accuracies. On the otherhand, the LSTM-based architecture is fairly successful on the auxiliary reconstruction task, but it is not sufficient for training on the main task to really take-off. As expected from the fact that the benchmark instantiates a binding problem that requires relational responding, our results hint at the fact that the ability to use memory towards deriving valuable relations between stimuli seen at different time-steps is primordial. Indeed, only the agent that has the ability to use its memory element towards recalling stimuli starts to perform at a better-than-chance level. Thus, the auxiliary reconstruction loss is an important element to drive some success on the task, but it is also clearly not sufficient, and the rather poor results that we achieved using these baseline agents indicates that new inductive biases must be investigated to be able to solve the problem posed in this benchmark.

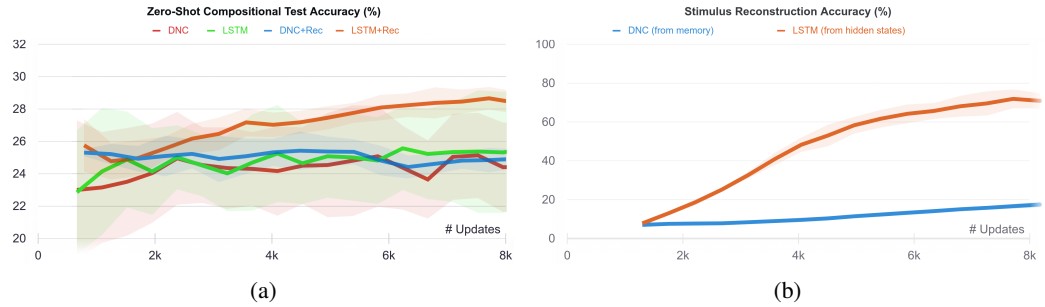

(a)  (b)

Figure 6: **(a):** 4-ways (3 distractors) zero-shot compositional test accuracies of different architectures. 5 seeds for architectures with DNC and LSTM, and 2 seeds for runs with DNC+Rec and LSTM+Rec, where the auxiliary reconstruction loss is used. **(b):** Stimulus reconstruction accuracies for the archiecture augmented with the auxiliary reconstruction task. Accuracies are computed on binary values corresponding to each stimulus' latent dimension's reconstructed value being close enough to the ground truth value, with a threshold of $0.05$.

# E  BROADER IMPACT

No technology is safe from being used for malicious purposes, which equally applies to our research. However, aiming to develop artificial agents that relies on the same symbolic behaviours and the same social assumptions (e.g. using CLBs) than human beings is aiming to reduce misunderstanding between human and machines. Thus, the current work is targeting benevolent applications. Subsequent works around the benchmark that we propose are prompted to focus on emerging protocols in general (not just posdis-compositional languages), while still aiming to provide a better understanding of artificial agent's symbolic behaviour biases and differences, especially when compared to human beings, thus aiming to guard against possible misunderstandings and misaligned behaviours.

Table 5: Hyper-parameters values used in R2D2, with LSTM or DNC as the core memory module. All missing parameters follow the ones in Ape-X (Horgan et al., 2018).

| R2D2 | |
|---|---|
| Number of actors | 32 |
| Actor parameter update interval | 1 environment step |
| Sequence unroll length | 20 |
| Sequence length overlap | 10 |
| Sequence burn-in length | 10 |
| N-steps return | 3 |
| Replay buffer size | $5 \times 10^4$ observations |
| Priority exponent | 0.9 |
| Importance sampling exponent | 0.6 |
| Discount $\gamma$ | 0.997 |
| Minibatch size | 32 |
| Optimizer | Adam (Kingma and Ba, 2014) |
| Optimizer settings | learning rate $= 6.25 \times 10^{-5}$, $\epsilon = 10^{-12}$ |
| Target network update interval | 2500 updates |
| Value function rescaling | None |

| Core Memory Module | | | |
|---|---|---|---|
| **LSTM (Hochreiter and Schmidhuber, 1997)** | | **DNC (Graves et al., 2016)** | |
| Number of layers | 2 | LSTM-controller settings | 2 hidden layers of size 128 |
| Hidden layer size | 256, 128 | Memory settings | 128 slots of size 32 |
| Activation function | ReLU | Read/write heads | 2 reading ; 1 writing |

