# OpenReview forum: "Meta-Referential Games to Learn Compositional Learning Behaviours"
_ICLR.cc/2024/Conference — Submitted to ICLR 2024_

### Official Review · Reviewer_CPjL · 2023-10-27

**Soundness:** 3 good
**Presentation:** 2 fair
**Contribution:** 3 good
**Rating:** 6
**Confidence:** 3

**Summary:**

The authors propose a benchmark for studying the ability of learning agents (in particular, multiagent RL learners) to learn compositional learning behaviors. The benchmark uses a meta-learning variant of referential games to instantiate this idea. The authors propose a "symbolic continuous stimulus" (SCS) representation to encode the semantic symbolic information in a domain-agnostic way, and then construct the datasets by drawing samples directly in this SCS space. The experimental evaluation shows that current approaches struggle to learn compositional learning behaviors.

**Strengths:**

######## Strengths ########
- The overview of the problems of systematicity/compositionality, lingustic compositionality, and compositionality of Sec 2 is valuable and interesting
- The problem of compositionality and compositional generalization is of interest to a large portion of the AI/ML community. Benchmarks in this direction are potentially highly impactful
- The experimental evaluation appears to be complete and useful (though some discussion is missing)

**Weaknesses:**

######## Weaknesses ########
- The description of the SCS is convoluted and hard to follow
- The overall evaluation protocol of the meta-referential games is not sufficiently clear

######## Recommendation ########

I recommend accepting the paper. The technical quality of the submission is high, the problem is of interest, and the benchmarking results demonstrate that existing methods struggle to solve the benchmark. I do have several suggestions for improvement which I hope the authors take.

######## Arguments ########

The main technical contribution of the paper is the problem formulation of meta-referential games and a synthetic benchmark that studies the setting. The idea is that, given a sufficient number of systematic generalization training problems, the listener/speaker agents should be able to learn a compositional learning behavior, such that they can generalize compositionally _in a new systematic generalization problem_. One additional technical contribution is the SCS, which is a domain-agnostic representation of a symbolic space. Unlike one-hot encodings, whose size depends on the number of values that each dimension can take, the SCS has a fixed size given a chosen dimensionality. For the benchmark, this implies that the different "tasks" can use varying semantic structures and the agents should still be able to meta-learn a compositional behavior.

I also appreciate the discussion of systematicity and disentanglement, though I have some comments/questions about that.

I have a few suggestions for improvement, which I think are necessary in order for the paper to be a complete technical contribution, which I summarize below:

- Details of the SCS
    - It's unclear what the tuple (d(i))_i... means. The authors then say that the "shape of a stimulus ... is a vector over [-1,+1]^N_dim". Is the shape a vector or is the representation a vector? If the vector is over [-1,+1] on every input, where does the d(i) the tuples factor in? The authors themselves state that the shape doesn't depend on the d(i)'s.
    - The later description says l(i) \in [1; d(i)] -- what does [1; d(i)] mean? is it the same as [1, dim(i)]? It seems that the authors might be using the two notations interchangeably
    - My understanding is that for every dimension i, l(i) picks an "index" from 1 to d(i), which is precisely the value of the stimulus at dimension i. Then, a Gaussian is sampled around that index with a small enough variance such that all samples fall near l(i) and are not confused with l(i)-1 or l(i)+1. If this is the case (which I think Fig. 3 confirms), the authors should attempt to make their textual description a bit clearer. As it stands, it is a bit convoluted.
    - The authors should carefully incorporate the answers to these questions and a cleaner explanation of the SCS in text.
- Evaluation protocol of the meta-RGs
    - My understanding of the first few lines is that generating "differently semantically structured" spaces is akin to generating many SCAN datasets. So each generated space is 1 SCAN dataset, and our goal will be to meta-learn a strategy that enables solving the ZSCT of a new SCAN dataset?
    - "a meta-referential game is composed of two phases" -- I'm confused by this. Isn't each RG itself composed of two phases, and the meta-RG a wrapping process that presents the two agents with many such RGs?
    - The authors put considerable efforts toward explaining the overall evaluation/training process, but it still doesn't appear to come through clearly. There are RGs and meta-RGs, shots and episodes. Each shot is a series of RGs. It is unclear exactly how all these pieces interact. I think the manuscript would leverage from one algorithm block that summarizes the overall process. For example:
```
Algo: Meta-RG evaluation process

    Meta-training phase:
    for episode in NumberOfEpsiodes // loop over tasks=episodes
        draw semantic structure
        for shot in NumberOfShots   // loop over ...
            draw component values
            for RG in ...
                draw stimulus
                ...
    Meta-testing phase:
    freeze speaker
    ...
```
    - The textual description is just too complex to come across clearly. Having an algorithmic description (and relying on it by referencing it in the textual description) might make things a lot clearer.
    - But overall, my understanding is that the agent faces a set of meta-training settings, each of which fixes one symbolic space and consists of many training RGs and zero-shot RGs. Then the agent faces meta-testing RGs, which presumably have little data?

**Questions:**

######## Additional feedback ########

The following points are provided as feedback to hopefully help better shape the submitted manuscript, but did not impact my recommendation in a major way.

Intro
- I'm not really sure I follow how the authors' view of online/offline relates to the RL view

Sec 2
- Fig. 1 -- why does the receiver also observe the state? Is it just a "noisy" version of the state w distractor stimuli?
- My understanding: the sender receives 1 input and communicates (potentially back-and-forth) with the listener, who additionally receives a set of inputs (potentially including the speaker's input or the same "object"). The task is for the receiver to determine, given messages from the sender, whether any of its observed stimuli match the speaker's. Some of this isn't explicitly stated, so it required looking at the figure. If there is such a 1-sentence explanation, I encourage the authors to include it at the beginning of their explanation before diving into the specific properties/variations.
- This section is a perhaps too philosophical discussion of the relations between disentanglement and compositionality, but I don't think that's necessarily a bad thing

Sec 3
- Authors state that in step N+2 the listener observes the input of the listener "rather than an object-centric samples with the same semantic meaning" --- but according to the definition, it's not _always_ the same semantic meaning, right? The game is to determine precisely whether the meaning is the same?
- "we propose a rule-based speaker" -- At this point, it seems that the only learning agent is the listener. But then (in Sec 4) the authors apparently clarify that this is only an ablative test to see how well the listener can learn CLBs given a fixed (linguistically compositional) speaker. This should be either omitted from this section or stated more clearly

Sec 3.2
- Vocabulary permutation: I wonder if it would be possible to construct a different stimulus representation that _doesn't_ require permutation to guarantee no cheating. Any insight from the authors on this? (In an ideal world, we would get a proof that no such representation exists, but an intuitive description of why that's difficult would also be valuable.)

Sec 4
- The authors report only results of the test/zero-shot performance. While this is the metric of interest, I wonder if it's possible, because of the difficulty of RL/MARL training, that even training performance is low? That would conflate the standard RL issues witht he issues of CLB.

Sec 4.1
- How is EoA measured? What about topsim/posdis/bosdis? What values should we expect for them? Is higher or lower better?
- Generally, I would expect a discussion that goes beyond just the zero-shot accuracy

Typos/style/grammar
- Fig. 1 (and others): authors should use a vector version of the image, not PNG or JPEG -- the size is small and zooming in blurs all letters/symbols
- Sec 2, "Compositionality..." -- "...the work ofHupkes et al. (2019)" --> missing space
- Sec 2, "Compositionality..." -- "... related contents"(Fodor et al., 1988)." --> missing space
- Sec 2, "Compositionality..." -- topographic similarity (topsim) vs. posdis (positional disentanglement) -- maintain consistency of abbreviations and parentheses
- Sec 2, "Compositionality..." -- I was initially confused by "and interchangeably compositional behaviors and systematicity..." because I thought you would use either of those two interchangeably with "linguistic compositionality". It would be clearer to write "and compositional behaviors and systematicity interchangeably to ..."
- Once the authors define the RG acronym, they should avoid going back and forth between RG and referential game
- Sec 3 -- "Figure 4(left)" --> missing space
- Sec 3.1 -- "relies on gaussian kernels" --> capitalize Gaussian (throughout the manuscript)
- Sec 3.1 -- "Figure 4(right)" --> missing space
- Sec 3.2 -- "an meta-referential game" --> "a meta..."
- "we bring the readers attention on" --> "we bring the reader's attention to"

---

> ### Author Response · Authors · 2023-11-20
> **Rebuttal (1/4)**
>
> Thank you for your very detailed and thorough review and feedback. We address them below.
> Please let us know if our replies and proposed changes are satisfactory and whether they contribute ton increasing your rating of our paper.
>
> ## Regarding 'some discussion [being] missing' :
>
> Following the feedback from Reviewer Hoy7, we clarified the experiments section and added a Discussion subsection to try to shore up our arguments.
> Please refer to our rebuttal to Hoy7 for further details, and do let us know whether you see any other ways to improve the paper on that end.
>
> ## Regarding Weaknesses in Description of the SCS :
>
> Thank you for your feedback and suggestions for improvement.
>
> We validate your understanding of the description of the SCS and include answers to your questions into the paper:
>
> 1. The representation is indeed the one that is a vector, not the shape. We also clarify how the d(i)'s factor in our reformulation, as follows:
>
>     "Stimuli in the SCS representation are vectors sampled from the continuous space $[-1,+1]^{N_{dim}}$. In comparison, stimuli in the OHE/MHE representation are vectors from the discrete space $\{0,1\}^{d_{OHE}}$ where $d_{OHE} = \Sigma_{i=1}^{N_{dim}} d(i)$ depends on the $d(i)$'s.Note that SCS-represented stimuli have a shape that does not depend on the $d(i)$'s values, this is the \textit{shape invariance property} of the SCS representation (see Figure~\ref{fig:posdis-messages+SCS-vs-OHE_rep}(bottom) for an illustration). In the SCS representation, the $d(i)$'s do not shape the stimuli but only the semantic structure, i.e. representation and semantics are disentangled from each other. The $d(i)$'s shape the semantic by enforcing, for each factor dimension $i$, a partitionaing of the $[-1,+1]$ range into $d(i)$ value sections. Each partition corresponds to one of the $d(i)$ symbolic values available on the $i$-th factor dimension. Having explained how to build the SCS representation sampling space, we now describe how to sample stimuli from it. It starts with instantiating a specific latent meaning/symbol, embodied by latent values $l(i)$ on each factor dimension $i$, such that $l(i)\in [1;d(i)]$. Then, the $i$-th entry of the stimulus is populated with a sample from a corresponding Gaussian distribution over the $l(i)$-th partition of the $[-1,+1$ range."
>
>
> 2. We emphasised that "$d(i)$ is the number of possible symbolic values for each latent/factor dimension $i$" in order to make it easier to understand the range "[1,d(i)] as the range from which latent/instantiated symbolic value l(i) is sampled, since there are d(i) such values.
>
>
> Please let us know if these changes improves the paper and whether you can see any other way for us to make this part of the paper easier to understand.
>
>
> ## Regarding Weaknesses in Description of the Evaluation Protocol :
>
> Thank you for your detailed feedback, we have addressed some of it while answering to Reviewer qdfE, so we invite you to read our rebuttal and let us know whether our proposed changes are aligned with your expectations too.
> In the meantime, we are still in the process of trying to address your feedback, but we thought it could be worth to start the conversation already in order to make sure that we are going in the right directions, as the end of the authors-reviewers discussion period is approaching fast...

---

> > ### Comment · Reviewer_CPjL · 2023-11-21
> > **Thank you for the partial response**
> >
> > I thank the authors for uploading this partial response. Indeed, these clarifications go a long way toward making the paper stronger. I do not have any further questions, pending the authors' final response. I will wait to receive all information and then consider the response as a whole along with the other reviews to determine whether a change in score is warranted.

---

> ### Author Response · Authors · 2023-11-22
> **Rebuttal (2/4)**
>
> Thank you for your acknowledgement of our clarifications so far being instrumental in making the paper stronger.
> We are providing further replies and proposed changes below :
>
>
> > So each generated space is 1 SCAN dataset, and our goal will be to meta-learn a strategy that enables solving the ZSCT of a new SCAN dataset?
>
> Your understanding is right. We would add a nuance in the fact each generated space is 1 domain-agnostic and proper-BP-instantiating SCAN dataset.
>
> Indeed, SCAN is on a specificly textual modality, and it does not instantiate a BP because each SCAN stimulus on its own clearly reveal the 'symbolic values' that they instantiate (which are words/tokens).
>
> > "a meta-referential game is composed of two phases" --I'm confused by this. Isn't each RG itself composed of two phases, and the meta-RG a wrapping process that presents the two agents with many such RGs?
>
> The meta-RG is indeed a wrapping process that presents the two agents with many RGs, but the first RGs have a supporting role for the agents to learn about the symbolic space that they are dealing with on the current episode, and the last RGs are meant for us to evaluate the agents on their ability to solve ZSCTs in this new context, i.e. ability to perform CLBs.
>
> Thus, we clearly mark a distinction between the supporting phase and the querying phase based on which type of stimulus is presented to the agents.
>
> Regarding your comment that common RGs themself are also 'composed of two phases', we are not sure what you are referring to?
>
> - If you mean to refer to the training and testing phase, then we mean to argue that those have more to do with the supervised learning loop in which common RGs are usually instantiated. Please refer to Algorithm 4 for clarification.
>
> - If you mean to refer to the first phase of a RG as being when the speaker operates, and the second phase as being when the listener operates, then we acknowledge that this is a possible framing of a RG, but we actually abide by a different framing which is based on communication rounds. Please refer to Denamganai et al., 2020a, or Algorithm 3 in Appendix A which details how a RG is defined in our framework.
>
> > The authors put considerable eorts toward explaining the overall evaluation/training process, but it still doesn't appear to come through clearly. There are RGs and meta-RGs, shots and episodes. Each shot is a series of RGs. It is unclear exactly how all these pieces interact. I think the manuscript would leverage from one algorithm block that summarizes the overall process.
>
> Thank you for your advice of adding an algorithm block. We have added 5 of those in order to try to explain in as many details as necessary the nuances that are involved.
>
> Throughout Section 3, we have added references to the different lines of those algorithm in order to provide landmarks to the readers.
>
> > But overall, my understanding is that the agent faces a set of meta-training settings, e ach of which fixes one symbolic space and consists of many training RGs and zero-shot RGs. Then the agent faces meta-testing RGs, which presumably have little data?
>
> We mean to highlight a misunderstanding here: the meta-training and meta-testing settings are maybe misleading naming :
>
> In a supervised learning loop, what differentiates training from testing settings are the data being used in each setting, having defined training and testing splits of the dataset.
>
> But, in the case of meta-learning with a distribution of task/dataset, each new episode that is generated by a new random seed (achieved by having the explicitly keeping a counter that is increasead by one at each reset of the environment, and this new counter value is used to seed the next environment) consist of never-before-trained-on data, therefore it is given that any new episode can be considered a testing-purpose episode.
>
> Thus, our algorithm 5 is not split around a meta-training and a meta-testing phase, but rather solely around meta-RL episodes and we report the ZSCT accuracy on each episode as our testing performance metric because each new episode involves novel data that can be treated as test before the agents later use that data for training.
>
> It does not invalidate our testing process since this data will never be seen by the agents again since the random seed will keep on increasing.

---

> ### Author Response · Authors · 2023-11-22
> **Rebuttal (3/4)**
>
> > My understanding: the sender receives 1 input and communicates (potentially back-and-forth) with the listener, who additionally receives a set of inputs (potentially including the speaker's input or the same "object"). The task is for the receiver to determine, given messages from the sender, whether any of its observed stimuli match the speaker's. Some of this isn't explicitly stated, so it required looking at the gure. If there is such a 1-sentence explanation, I encourage the authors to include it at the beginning of their explanation before diving into the specic properties/variations.
>
> Thank you for your feedback about the need for a 1-sentence-like explanation around RGs, we have included the following sentences based on yours :
>
> "In short, the speaker receives a stimulus and communicates with the listener (up to $N$ back-and-forth using messages of at most $L$ tokens each), who additionally receives a set of $K+1$ stimuli (potentially including a semantically-similar stimulus as the speaker).
>
> The task is for the listener to determine, given communication with the speaker, whether any of its observed stimuli match the speaker's."
>
> > Authors state that in step N+2 the listener observes the input of the listener "rather than an object-centric samples with the same semantic meaning" ---but according to the denition, it's not always the same semantic meaning, right? The game is to determine precisely whether the meaning is the same?
>
> Yes, your understanding is correct, and we realise that our sentence makes it ambiguous. Thank you for your feedback, we reformulate this sentence as follows:
> "Next, step $N+2$ is intended to provide feedback to the listener agent as its observation is replaced with the speaker's observation (cf. line $12$ and $18$ in Alg.5). Note that this is the exact stimulus that the speaker has been observing, rather than a **possible** object-centric sample."
>
> > "we propose a rule-based speaker" --At this point, it seems that the only learning agent is the listener. But then (in Sec 4) the authors apparently clarify that this is only an ablative test to see how well the listener can learn CLBs given a xed (linguistically compositional) speaker. This should be either omitted from this section or stated more clearly
>
> Thank you for your advice, we are now omitting this sentence from this section and only unveiling it during the Experiment section.
>
> > Vocabulary permutation: I wonder if it would be possible to construct a dierent stimulus representation that doesn't require permutation to guarantee no cheating. Any insight from the authors on this? (In an ideal world, we would get a proof that no such representation exists, but an intuitive description of why that's dicult would also be valuable.)
>
> The vocabulary permutation scheme is not only a trick to prevent the the agents from building a cheating language, it is also a legacy feature from the Emergent Communication field building towards AI's ability to communicate with 'strangers'/novel partners.
>
> We ~~(will)~~ discuss this matter further in a subsequent response to Reviewer Hoy7's reply to our rebuttal.
>
> Following your prompt to build a symbolic space that would inherently guard against the emergence of a cheating language, we would propose **presenting** stimuli on a different range than the $[-1, +1]$ range that is currently used, for instance by applying an affine transformation (i.e. offset and rescaling) to the whole representation space before showing stimulus to the agents.
>
> Indeed, by applying a different randomly sampled affine transformation at each episode, we would prevent the agents from being able to expect the stimulus values to be bounded (with a max and min) or to have a minimal precision requirement.
>
> Thus, the agents would not be able to build an analog-to-digital conversion-inspired language since the two properties of the data to make such a conversion would be lacking.
>
> We have not experimented with it yet, but we are looking into this in order to relax the 'communication with stranger' aspect of the benchmark in a subsequent work, where the constructivity aspects can be studied with even less confounding variables.
>
> We might be able to present some preliminary results of that context in the final version of the paper, should it be accepted for publication, but the current author-reviewer discussion period does not give us enough time to run all the necessary experiments, and we also think that the paper is already 'meaty'-enough as it is...
> Please let us know your though on the matter.

---

> > ### Author Response · Authors · 2023-11-22
> > **Rebuttal (4/4)**
> >
> > > The authors report only results of the test/zero-shot performance. While this is the metric of interest, I wonder if it's possible, because of the diculty of RL/MARL training, that even training performance is low? That would conate the standard RL issues witht he issues of CLB.
> >
> > As discussed above with regards to the algorithms, we mean to clarify that given the meta-learning context and our environment seeding approach relying on an always-increasing counter, each new episode instantiates a novel task, making it a valid testing episode, **at the time of harvesting**.
> >
> > Once over, the agent is trained on the episode, as the trajectory is added to the experience replay buffer of our RL agent.
> >
> > The related symbolic space is never resampled during harvesting though, thanks to the determinism of using pseudo-random generation for all aspects of the benchmark.
> >
> > Thus, we only report testing measures because all new episode that we harvest are testing episodes, until the moment they are used for training.
> >
> > We appreciate the idea proposed by the reviewer though, and could see the value in reporting a validation-like accuracy measure, that would be derived from a recurring set of environments, throughout the meta-training process.
> >
> > Please let us know whether you think that would be valuable ?
> >
> > > How is EoA measured? What about topsim/posdis/bosdis? What values should we expect for them? Is higher or lower better? Generally, I would expect a discussion that goes beyond just the zero-shot accuracy
> >
> > We have provided further details with regards to how EoA is measured in the related Evaluation paragraph, and made it clearer with an upward arrow in Table 2 that all metrics are higher-is-better.
> >
> > We have added a Discussion subsection in order to carefully reframe our results, following feedback from Reviewer Hoy7.
> >
> > Please let us know if you would expect anymore details or discussion.
> >
> > This concludes our rebuttal, we thank you again for your time and very detailed feedback.
> > We are looking forward to any of your subsequent comments towards possibly improving further the quality of our paper.
> >
> > In the meantime, we focus on trying to integrate into the Discussion & Related Works section more of the arguments we made towards our benchmark and CLBs to be impactful, in our reply to Reviewer Hoy7.

---

### Official Review · Reviewer_Hoy7 · 2023-10-28

**Soundness:** 2 fair
**Presentation:** 2 fair
**Contribution:** 3 good
**Rating:** 6
**Confidence:** 3

**Summary:**

This paper introduces a emergent communication benchmark/game called the
meta-referential game.  It is based on the familiar referential game from the
EC literature but is posed in a meta-learning framework which requires the
agents to establish communicative conventions within an episode of iterated
referential games.  Such a game requires agents to learn to dynamically acquire
language (i.e., over the course of an episode) rather than simply learn
a static mapping as happens in the standard referential game.  Empirical
analysis adds some context to how baseline approaches fare in different
hyperparameter settings of the benchmark/game.

**Strengths:**

- (major) The benchmark introduces this concepts of receptivity and
  constructivity (i.e., the ability to establish linguistic conventions within
  an episode) into emergent language.  These are indeed present in human
  language behavior but not often (if at all) discussed in the context of
  emergent language.
- (major) The meta-referential game is largely an appropriate extension to the
  referential game which introduces the necessary concepts for intra-episode
  learning without making too many changes (i.e., which could introduce too
  many confounding factors).

**Weaknesses:**

- (major) The empirical results are difficult to interpret in a meaningful way
  since the main ones are negative, and there are not many clear trend in the
  rest.  While the primary contribution of this paper is the benchmark, it is
  tough to see whether or not it will be of practical use based on the
  empirical results presented.
- (minor) On the level of clarity, the paper uses a lot of jargon that is a bit
  distracting.  Even if most of these terms are defined, it makes for
  a difficult read.  This could just be a background mismatch is I come from an
  NLP/RL/emergent communication background.  Technical terms do make things
  clearer and more precise in moderation, but when they proliferate, it
  obscures instead.  Some terms I'm referring to:
  - binging problem ("binding" itself is never actually defined, I think)
  - compositional learning behavior
  - reflexivity and constructivity
  - object-centric versus stimulus-centric
  - Chaa-RSC and Hill-RSC
  - shape invariance property and semantically structured symbolic spaces
  - Symbolic Continuous Stimulus
- (minor) The "Symbolic Continuous Stimulus" seems to be a bit more complicated
  than it needs to be; namely with the many layers of sampling (i.e., the
  number of partition, the size of the partitions, the parameters of the
  Gaussian, then the Gaussian itself) that just create the data distribution.
  I do see how some of this is necessary to prevent confounding factors, but
  I think preemptively ramping the complexity of the benchmark when it is not
  even clear that current models can do much better than random chance might
  not be the right move.

**Questions:**

What do the empirical results show?  And how do these findings support the
benchmark?

### Misc comments

- It is a little confusing with all of the parameters "shots", "steps",
  "games", "meta games" (although I understand why these are necessary).  To
  alleviate this somewhat, it might be worthwhile to include a table that just
  lists a sample set of interactions, observations, etc. in a table format
  (which could definitely could be hand written/not real) to give a sense of
  what the parameters correspond to.

- Page 1
  - "In this work, we will primarily...": don't use a "respectively" sentence structure here, it makes it very difficult to read this important sentence.
- Page 2
  - The definition of the binding problem is not clear at all since what "binding" actually is never defined -- it's somewhat circular
  - "(Lazaridou and Baroni, 2020)" - use `\citet`
- Page 4
  - "semantical" -> "semantic"
  - "S2B" -> "SB2"? The postfix two usually represents a superscript.
- Page 5
  - "segregated" -> "segregate"
  - First paragraph of Sec 3.1 was difficult to understand on the first
    read-through.  It was clearer reading it a second time (after reading
    through the whole paper), and think the reason is because SCS is not
    discussed in detail until after this paragraph despite the fact that the
    nature of SCS is important to understanding this paragraph.  This is
    coupled with the fact that the "binding problem" is never full defined
    (i.e., what "binding" is in the first place).
  - Figure 2: what is the difference between the "object-centric target
    stimulus" and the "target stimulus"?
  - "but not larger than the size of the partition section it should fit in":
    not possible since Gaussian distributions have infinite support for any
    non-zero standard deviation.  Does SCS use rejection sampling to ensure
    that out-of-bound samples do not get passed along?
  - maybe just have uniform sampling from the partitions or just have Gaussian
    sampling from a list of means
  - how are the spaces partitioned?
  - What is the structure of a semantic space, just the layout of partitions?
- Page 6
  - What is the "shape invariance property"?
  - "an meta-referential" -> "a meta-referential"
  - Figure 2: maybe referring to a "referential game" as a "round" would be
    clearer
  - "attention on the fact" -> "attention to the fact"
  - Not clear what a "random permutation of the vocabulary symbols" means.
- Page 7
  - 4.Agent Architecture - It would be best to at least give a 2-sentence
    summary of the arch.
  - Adding this auxiliary loss definitely merits discussion in the overall
    context of the benchmark, i.e., how it might affect what the benchmark
    would and would not show.
  - "make emerge a new language": rephrase; maybe "invent a new language"?
  - "resolution approach": rephrase
  - "K = 0": Seems out of place to parameterize a value when it is just going
    to result in a binary task.
  - "goads us to think" -> "leads us to think"
  - Sec 4.2.1 - It is difficult to tell here if the results are showing
    anything significant.

---

> ### Author Response · Authors · 2023-11-16
> **Rebuttal (1/1)**
>
> Thank you for your very detailed review and feedback, it is greatly appreciated.
> We address them below:
>
> ## Major Weakness Feedback :
>
> Thank you for your feedback on the difficulty to interpret the paper's empirical results and the lack of trends in our experiment section.
> We address it as follows:
>
> Firstly, we adapt the experiment's  titles to highlight the main takeaways and make it easier for the reader.
>
> Secondly, we re-framed the paragraphs introducing each experiments in order to make it clearer what is the trend/narrative in our experiments.
> We used this opportunity to better emphasise the two main aspects of the problem, to wit the receptivity and constructivity aspects of CLBs, following your highlighting of those elements as major strengths of the paper.
>
> Then, we simplified the experiments of the main text, by pushing into the appendices our investigation of the impacts of memory and sampling budget on performance.
>
> The space that we gained was used to include a Discussion subsection that summaries and re-frame the results in order to clarify the practical use of the benchmark.
> Please refer to this as our answers to your two explicit questions on top of the  'Questions' section of your review.
>
> Please let us know if these changes enhance your appreciation of the paper and/or whether you have any other suggestions.
>
> ## Minor Weaknesses Feedback:
>
> Thank you for your feedback, we have tried to reduce our usage of jargon and acronyms while abiding by the page limit.
> Please let us know if you see any specific approach to further improve on that end.
>
> ## Misc Comments :
>
> Thank you for your miscellaneous comments, we address as much as possible in the following:
>
> - We reframed the critical sentence that previously used a 'respectively' framework.
>
> - We added [/re-use] as a sinonymous expression to 'bind' in our citation of Greff et al., 2020, and elaborated further what it means to solve the binding problem in the following terms: 'Solving the BP instantiated in such a context, i.e. re-using previously-acquired information in ways that serve the current situation, [...]'
>
> - We added to the main text the following sentence in order to clarify what is the relationship between the 'target stimulus' and the 'object-centric target stimulus' : `The adjective `object-centric' is used to qaulify a stimulus that is different from another but actually present the same meaning (e.g. same object, but seen under a different viewpoint).'
>
> - SCS does not use rejection sampling to ensure that out-of-bound sampled do not get passed along, but the standard deviation is actually sampled from a range that is tighter than the actual available space, so we assume this to be sufficient.
>
> - The structure of a semantic space is indeed the layout of partitions, if by 'layout' you refer to the parameters of the Gaussian kernels.
>
> - Regarding the 'shape invariance property', please refer to our answer to Reviewer 3drE. We address the feedback by adding a figure highlighting the shape invariance property (Figure 4 (bottom)).
>
>
> Once again, we thank you for your review and feedback. Please let us know if our replies and made changes increase your ratings of our paper, and whether, in light of other reviews and rebuttals, you think of anything new to improve the paper.

---

> > ### Comment · Reviewer_Hoy7 · 2023-11-21
> > **Response to rebuttal**
> >
> > I appreciate the author's response, and I think the paper has been improved through that.
> > In light of this, I will raise my rating of "Presentation" from a 2 to 3.
> > I will ultimately decide not to raise my overall rating of the paper above 6.
> > I do think the paper is making _a_ contribution insofar as it is introducing a benchmark for compositional learning behaviors, which are themselves interesting.
> > While they may be interesting, I am not convinced that they are important in a practical way.
> > To illustrate what I mean, I'll start with this quote from the paper:
> >
> > > [The] results validate the need for our benchmark and they highlight that
> > > our efforts should be focused on constructivity aspects of CLBs.
> >
> > I think the importance of CLBs is apparent from a conceptual level, but what the experiments and results do not show is that CLBs (and specifically this benchmark) are particularly important to the development of emergent communication as a whole.
> > That is, I am not convinced that this benchmark would see widespread use in EC research as it progresses towards its own practical goals.
> > I believe it has some use, hence my rating of a 6, but I am not sure it has a major use, which is what keeps me from championing this paper with an 8.

---

> ### Author Response · Authors · 2023-11-22
> **On the Potential Widespred Use of our benchmark and CLBs (1/3)**
>
> Thank you for your response and acknowledgement of the presentation of the paper having improved.
> Thank you for your feedback regarding your doubts that CLBs and our benchmark at large can be interesting from a practical viewpoint and that they might not have widespread use.
> We would like to discuss this point further and attempt to provide different arguments in order to argue that they might get widespread use, and especially not solely in the EC field but for the whole of the AI field :
>
> ## Argument 1 :
> Our benchmark proposes a synthesis between Chaa-RSC and Hill-RSC towards identifying conditions/causes that lead to the emergence of systematicity in connectionnist architectures (neural network-based architectures - NNs).
> We stress that Chaa-RSC stems from the work of Chaabouni et al. [1] which, with its 96 citations over the last 2 years, represents a very important work in the Emergent Communication subfield and beyond, as the number of citations highlights.
> Our benchmark provised an avenue for related research questions to be addressed by exposing relevant parameters and also providing an opening to the related hypotheses made by Hill et al [2], coming from the language grounding and embodied AI fields.
> Hill et al.[2] has collected 93 citations so far, thus showing that it was an important contribution.
> Since our benchmark builds a bridge between these two (too-often-separated) subfields of NLP, by linking two papers that have garnered a more than fair interest, we argue that our benchmark is bound to be a fertile ground for subsequent and related research stemming from both the subfields of EC, language grounding, and Embodied AI.
>
> [1] : Chaabouni, Rahma, et al. "Compositionality and Generalization In Emergent Languages." _Proceedings of the 58th Annual Meeting of the Association for Computational Linguistics_. 2020.
>
> [2] : Hill, Felix, et al. "Environmental drivers of systematicity and generalization in a situated agent." _International Conference on Learning Representations_. 2019.

---

> > ### Author Response · Authors · 2023-11-22
> > **On the Potential Widespred Use of our benchmark and CLBs (2/3)**
> >
> > ## Argument 2 :
> > We would like to highlight how the subfield of Emergent Communication has been building in the direction of our benchmark since the 2020 edition of the Emergent Communication Workshop at NeurIPS whose theme was 'Talking to Strangers'[1].
> > Indeed, the extended dimension at that time was towards zero-shot communicating with novel partners _(about known data)_.
> > Among the accepted papers can be found the work  Cope et al.[2] which has been addressing how to instantiate this novel-partner communication setting with some graduation and a more principled way by proposing to simulate speaking with a  'stranger' as speaking with a known partner when the vocabulary is permuted (thanks to the vocabulary permutation scheme).
> > Then during the subsequent edition of the Emergent Communication workshop held at ICLR in 2022 [4], the them was 'New Frontiers' thus emphasising a will to find practical applications and synthesis opportunities with other fields.
> > Notably, the work of Cope et al.[5] proposed an interesting new framework for **cooperative language acquisition** for ad-hoc team play.
> > In our benchmark, we build over this trend in two ways:
> >
> > 1. we shore up the theoretical framework by incorporating concerns related to symbolic behaviours, and more specifically we divide and conquer the problem of speaking with strangers over the constructivity aspects and receptivity aspects of symbolic behaviours. This separation of concerns can also be found in the work Cope et al.[5]  in their 'Forward Problem (Signalling)' and 'Backward Problem (Listening)'. Our experiments and results advances the landpost by showing that our efforts towards enabling AI to interact with strangers should be focused on the constructivity aspects rather than the receptivity aspects, i.e. the 'Forward Problem' more than the 'Backward Problem'.
> > 2. we extend further the paradigm of emergent communication in the natural direction of talking with strangers **about novel data**, whereas accepted works at the NeurIPS 2020's workshop were limited to known data [2,3]. We do so thanks to our meta-RG extension which addresses AI abilities to zero-shot communicate with novel partners about novel data.
> >
> > With this build up that can be felt over different instances of the Emergent Communication workshop, we hope that you gain better hope that our benchmark would garner sufficiently-widespread use in the EC subfield alone.
> >
> > [1] : Emergent Communication Workshop at NeurIPS 2020 : https://sites.google.com/view/emecom2020/home
> >
> > [2] : Cope, Dylan, and Nandi Schoots. "Learning to communicate with strangers via channel randomisation methods." _arXiv preprint arXiv:2104.09557_ (2021). **Emergent Communication Workshop at NeurIPS 2020.**
> >
> > [3] : Bullard, Kalesha, et al. "Exploring zero-shot emergent communication in embodied multi-agent populations." _arXiv preprint arXiv:2010.15896_ (2020).**Emergent Communication Workshop at NeurIPS 2020.**
> >
> > [4] Emergent Communication Workshop at ICLR 2022 : https://sites.google.com/view/emecom2022/
> >
> > [5] : Cope, Dylan, and Peter McBurney. "Joining the Conversation: Towards Language Acquisition for Ad Hoc Team Play." **Emergent Communication Workshop at ICLR 2022**. 2022. (https://openreview.net/forum?id=SLqgf7ZCQbq)

---

> > > ### Author Response · Authors · 2023-11-22
> > > **On the Potential Widespred Use of our benchmark and CLBs (3/3)**
> > >
> > > ## Argument 3 :
> > > Compositionality (and here we mean compositional behaviours only (CBs)) is already a challenge that makes the field of Emergent Communication relevant to other fields of AI since systematic generalization (and value alignment) is probably what AI currently lacks the most.
> > > On top of this bridge from the Emergent Communication field to the rest of the AI community, our benchmark introduces into the Emergent Communication field another challenge that Greff et al.[1] highlights as critical for the AI communicaty, it is the challenges posed by the binding problem (BP).
> > > Thus, we argue that not only our benchmark is of importance to the Emergent Communication field towards being relevant to the rest of AI research, but our benchmark is also useful for the AI community that Greff et al.[1] references.
> > > Inded, our benchmark is the only one that addresses challenges of the BP in a principled way, and does so with the highest internal and external validity possible since our benchmark instantiates **domain-agnostic** BPs, which cannot be resolved by leveraging other aspects of the data.
> > >
> > > Finally, with the acknowledgement of CBs being already studied in the context of EC and valuable at large for the whole AI community, where do CLBs fit, if at all?
> > >
> > > CLBs have actually been at play in the work of Lake[2], Conklin et al.[3], and more recently (and also maybe more importantly, as a paper published in _Nature_) Lake and Baroni[4].
> > > Indeed, these works proposed meta-learning extensions of common settings (e.g. seq2seq in [2]) towards enabling systematicity in AI.
> > > But, on the contrary to our work, these works perform evaluation on CB-related benchmarks (SCAN, COGS) after using CLBs as a training tool only, whereas we propose a CLB-principled benchmark for evaluation of CLBs abilities themselves.
> > > This trend being fairly recent and only starting, we argue that our principled benchmark is very likely to find widespread usage in order to address research questions related to this trend.
> > >
> > > [1] : Greff, Klaus, Sjoerd Van Steenkiste, and Jürgen Schmidhuber. "On the binding problem in artificial neural networks." _arXiv preprint arXiv:2012.05208_ (2020).
> > >
> > > [2] : Lake, Brenden M. "Compositional generalization through meta sequence-to-sequence learning." _Advances in neural information processing systems_ 32 (2019).
> > >
> > > [3] : Conklin, Henry, et al. "Meta-learning to compositionally generalize." _arXiv preprint arXiv:2106.04252_ (2021).
> > >
> > > [4] : Lake, Brenden M., and Marco Baroni. "Human-like systematic generalization through a meta-learning neural network." _Nature_ (2023): 1-7.
> > >
> > > After detailing these three arguments, we would be very grateful if you were to keep on engaging into this conversation.
> > > We are in the process of trying to add parts of this argumentation to our Related Works section and could really use your feedback, and that of the other Reviewers, please.

---

> > > > ### Author Response · Authors · 2023-11-23
> > > > **Paper Revision : Including Argument 3**
> > > >
> > > > With the authors-reviewers discussion period ending soon, we have updated our paper towards integrating only Argument 3 into the Discussion & Related work section.
> > > >
> > > > We hope that this will improve your ratings of our paper, and we thank you again for your time and efforts towards prompting us to improve the paper.

---

### Official Review · Reviewer_qdfE · 2023-10-31

**Soundness:** 3 good
**Presentation:** 3 good
**Contribution:** 2 fair
**Rating:** 5
**Confidence:** 2

**Summary:**

This paper proposes the Symbolic Behaviour Benchmark (S2B) to evaluate compositional learning behaviors (CLBs), especially the domain-agnostic binding problem (BP) instantiated by Symbolic Continuous Stimulus (SCS) representation.
It proposes a framework of Meta-Referential Games, a meta-learning extension of referential games (RGs).
The baseline results and error analysis show it is a compelling challenge.
It helps to make artificial agents collaborate with humans.

**Strengths:**

- The benchmark evaluates compositional behavior and binding problems, which are important problems in artificial intelligence.

- It proposes the Symbolic Continuous Stimulus instead of using the one-hot or the multi-hot encoded schemes.

- It proposes the Meta-Referential Games framework, which extends common referential games.

**Weaknesses:**

The main concern is that the benchmark may lack novelty.
Compared with common referential games, the proposed benchmark has SCS stimuli representation and the meta-learning extension.

(1) **Is the selection of representation essential for the benchmark of compositional generalization?**

The SCS representation has the advantage over one-hot or multi-hot representation.
However, it might not be essentially very important for the game framework.
For compositional generalization, the core point is that the test data has new combinations of stimuli.

(2) **The Meta-Referential Game framework and common referential games seem to have a similar protocol, so why only one of them is meta-learning?**

In the Meta-Referential Game framework, a game (episode) has a training phase and a test phase.
Do common referential games also have these two phases?
If so, it seems strange to say the Meta-Referential Game framework is a "meta-learning " extension to common referential games.

In the proposed framework, the stimuli in test RGs are recombined in novel ways, different from common referential games. Still, this difference seems not related to whether it is a meta-learning framework or not.

**Questions:**

(3) Does the SCS still have the advantage when used in general compositional generalization problems? How about in i.i.d. problems?

(4) It might be more reader-friendly to increase the size of the figures or the font size in the figures.

---

> ### Author Response · Authors · 2023-11-13
> **Rebuttal (1/2)**
>
> Thank you for your review and feedback, we proceed to address you questions below, as well as a possible misunderstanding.
> Please let us know whether those answers (below) and proposed changes (cf. revised paper) are helpful in increasing your ratings of our paper.
>
> ## Regarding your main concern: lack of novelty :
>
> We mean to bring your attention on a possible misunderstanding:
> Your review states that our benchmark lacks novelty (first sentence of the weaknesses paragraph) and that it 'evaluates compositional behaviours' (CBs) (cf. first strength bullet).
>
> There are indeed many benchmark that **only** evaluates CBs, for instance:
> - SCAN [1] ;
> - gSCAN [2];
> - COGS [3];
>
> But, we mean to emphasise that our benchmark aims to evaluate compositional **learning** behaviours (CLBs).
> Despite the fact that 'CLB' is linguistically a very similar referring expression to 'CB', it is in fact a very different problem to tackle.
> The problem of learning CLBs does involve compositionality like that of learning CBs, but it adds an extra difficulty to the task:
> If we define CBs as "the ability to generalise from combinations of known, **trained-on** atomic components to novel re-combinations of those very same atomic components", then we can define CLBs as "the ability to generalise from a few combinations of never-before-seen atomic components to novel re-combinations of those very same atomic components'.
> CLBs involve a few-shot learning aspect that is not present in CB.
>
> Thus, CLBs have not been addressed in the AI literature so far, thus making it a novel problem to consider, as we attempted to detail in Section 5 Related Works.
>
> Please let us know if this clarifies any possible misunderstanding.
>
> We are re-writing the first and third paragraph of the introduction in order to shore up these aspects of our work, adding reference to [4] and more.
>
> We would like to propose to maybe rename CBs into Compositional Inference-only Behaviours (CIBs) ?
> If you think this is worth doing, please let us know?
>
>
> [1] : Lake, Brenden, and Marco Baroni. "Generalization without systematicity: On the compositional skills of sequence-to-sequence recurrent networks." International conference on machine learning. PMLR, 2018.
>
> [2] : Ruis, Laura, et al. "A benchmark for systematic generalization in grounded language understanding." Advances in neural information processing systems 33 (2020): 19861-19872.
>
> [3] : Kim, Najoung, and Tal Linzen. "COGS: A compositional generalization challenge based on semantic interpretation." arXiv preprint arXiv:2010.05465 (2020).
>
> [4] : Beck, Jacob, et al. "A survey of meta-reinforcement learning." arXiv preprint arXiv:2301.08028 (2023).
>
> ## Question 1: Is the selection of representation essential for the benchmark of compositional generalisation?
>
> In light of our previous clarifying of the possible misunderstanding, we reframe the question to be: 'is the selection of the representation essential for the benchmark of CLBs?'
>
> We have highlighted in the third paragraph of the introduction how CLBs involve the resolution of a binding problem (BP).
> In appendix D.1 we provide evidence that the OHE representation does not instantiate a BP, on the contrary to our proposed SCS representation.
> Thus, selection of the SCS representation is criticial for evaluation of CLBs (but not for CBs, as your initial question suggested).
>
> Should our paper be accepted, and therefore granted a 10th page of content, then we would propose to re-include appendix D.1 back into the main content.
> Please let us know if this is satisfying to address your feedback or whether you would like to suggest something else?
>
> ## Question 2: The Meta-Referential Game framework and common referential games seem to have a similar protocol, so why only one of them is meta-learning?
>
> We have included in Appendix algorithms to detail the differences between common RG, sitting inside a supervised learning loop, and our proposed meta-RG which sits inside a meta-RL loop.
>
> We agree with your assessment that the both of them have similar protocol, up to the extent of the following:
> - the type of data that is considered throughout the protocol ;
> - the position of the agents parameters' update step in the protocol ;
> - and, obviously, the type of skills that they involve: CLBs vs CBs (as emphasised above).
>
> Please let us know whether you would appreciate us to add anything more than those algorithms (and previously discussed matters) to address your feedback.

---

> > ### Author Response · Authors · 2023-11-13
> > **Rebuttal (2/2)**
> >
> > ## Question 3: Does the SCS representation still has the advantage when used in general compositional generalization problems? How about in i.i.d. problems?
> >
> > The concerns of the paper being CLBs, we did not wanted to risk further confusing the reader by introducing CB-focused experiments, but our initial inquiries shows that agents are able to deal with SCS representation (using a fixed symbolic space's semantic structure) similarly to OHE representation inside common RG. There seems to be no pros nor cons, when playing common RGs (in supervised learning contexts).
> >
> > ## Question/Feedback 4:
> >
> > Thank you for your feedback, we will attempt to increase the font in our figures (especially Figure 1).

---

> > > ### Comment · Reviewer_qdfE · 2023-11-20
> > >
> > > Thank you for the detailed explanations. I have the following comments.
> > >
> > > > Novelty
> > >
> > > The paper’s main topic is CLB, a compositional extension of the problem considered in common reference games (we may call it the learning behavior (LB) problem). By saying ”lack of novelty,” I mean it
> > > does not seem difficult to discover the CLB problem, given the LB problem. For a machine learning task,
> > > there is a straightforward way for compositional extension by reorganizing data according to atom (factor)
> > > combinations, also used in this work (LB to CLB).
> > >
> > > > SCS representation
> > >
> > > SCS representation is useful for the binding problem (BP) in the CLB problem. However, the (domain-
> > > agnostic) BP also seems important in common reference games (LB problems). So, the SCS representation
> > > seems not directly related to compositionality, the paper’s main topic. Instead, it is another dimension of
> > > extending common reference games.
> > >
> > > > Meta-learning
> > >
> > > It seems to me that the position of the agent parameter update is not the essential difference. Common
> > > reference games can also update parameters at the end of an episode (batch update).
> > >
> > > In Algorithm 5, training stimulus sampling does not depend on model parameter updates. So, all training
> > > stimuli can be sampled before starting training. More specifically, stimulus sampling (Line 6) and S-support
> > > update (Line 10) are not influenced by other steps (Line 7-9) in the loop. So, Line 6 and Line 10 can be
> > > in an independent loop inserted between Line 4 and Line 5. Given the training stimuli, the two algorithms
> > > (Algorithm 4 and 5) can share the rest of the process in an episode.

---

> ### Author Response · Authors · 2023-11-21
> **Response (1/3)**
>
> Thank you for your follow-up comments. We appreciate you taking the time to continue this discussion.
>
> ## Regarding Novelty:
>
> We are afraid that we still fear there is a misunderstanding: while we agree with your comment that CLBs do build upon **the behaviours that are required to perform well in the context of referential games** (referring to those behaviours as 'LBs', as you propose the naming), but we disagree on the stance that CLBs involve a 'compositional extension'.
> Indeed, compositionality, or compositional behaviours (CBs), are already studied in the context of referential games[3] and variants, as seen in the field of emergent communication at large [1,2].
> Rather, training learning agents to perform CLBs requires a **few-shot learning** adaptation (which is a type of meta-learning [4]), and therefore, we mean to argue that **CLBs involve a meta-learning extension to referential games**.
> We apologize if we did not make this distinction clear earlier.
>
> We hope that, in light of this clarification, the rest of the paper can be seen under a better light, and possibly help improve your ratings of our paper.
> Please let us know if you have any other proposition for us to try to make the paper better.
>
> [1] : Lazaridou, Angeliki, and Marco Baroni. "Emergent multi-agent communication in the deep learning era." _arXiv preprint arXiv:2006.02419_ (2020).
>
> [2] : Choi, Edward, Angeliki Lazaridou, and Nando de Freitas. "Compositional Obverter Communication Learning from Raw Visual Input." _International Conference on Learning Representations_. 2018.
>
> [3] : Chaabouni, Rahma, et al. "Compositionality and Generalization In Emergent Languages." _Proceedings of the 58th Annual Meeting of the Association for Computational Linguistics_. 2020.
>
> [4] : Beck, Jacob, et al. "A survey of meta-reinforcement learning." arXiv preprint arXiv:2301.08028 (2023).

---

> > ### Author Response · Authors · 2023-11-21
> > **Response (2/3)**
> >
> > ## Regarding the SCS Representation:
> >
> > > SCS representation is useful for the binding problem (BP) in the CLB problem. However, the (domain- agnostic) BP also seems important in common reference games (LB problems). So, the SCS representation seems not directly related to compositionality, the paper’s main topic. Instead, it is another dimension of extending common reference games.
> >
> > We mean to nuance that the topic of our paper is CLBs, which is different from what your usage of 'compositionality' alone seems to refer to.
> > In the revised introduction (as well as in our Related Works section), we use the expression 'compositional behaviours' (CBs) to refer to what you seem to be referring to when you employ the term 'compositionality' without further details.
> >
> > We have common ground on the idea that the SCS representation is useful to instantiate a proper binding problem (BP) and therefore necessary for the study of CLBs.
> > We agree that referential games involve studying a kind of BP, but we mean to emphasise how it is different from the BP involve in CLBs:
> > Common referential games (RG) take place in a supervised learning loop and the CBs that they involve is between **trained-on** atomic components and novel/**test-time** re-combinations of those very same component.
> > Therefore, the BP that matters and is instantiated during the supervised learning loop of common RGs occurs during the testing part of that loop, and it involves 'binding'/re-using information about the following:
> >  1. **trained-on** atomic component, which are no longer observable, but only recoverable from having been (hopefully) **stored in the weights/parameters of the model/agent** ;
> >  2. and the currently observed stimuli that involve a novel recombination of previously-trained-on atomic component.
> > Due to the reliance on **trained-on** atomic components, we would refer to this kind of BP as an **offline BP**.
> > On the otherhand, the kind of BP that occurs during the meta-RL loop of meta-RGs, which is the kind of BP involved in the context of learning CLBs, involve binding/re-using information about the following:
> >  1. **never-before-seen**/not-trained-on atomic components which are observed during the supporting phase of the meta-RG and progressively **stored in the memory component of the model/agent** (not in the weights/parameters, since no parameter update is involved in the middle of a meta-RG) ;
> >  2. and the currently observed stimuli that involve a novel recombination of the newly-encountered atomic component.
> > Due to the reliance on **never-before-seen** atomic components, we propose to refer to these kind of BPs as **online BP**.
> >
> > Eventually, with that nuance in mind, we argue that the SCS representation is necessary for the study of CLBs for (i) it alone, to our knowledge, can instantiate **online BP**, and (ii) the study of CLBs requires instantiation of online BPs.
> >
> > This nuance was not discussed in Greff et al., (2020), which, to our understanding, mainly discussed offline BPs.
> > As you can appreciate, this nuance demands careful examination of a kind that the available space in the paper does not allow.
> > Thus, we decided to omit it from the paper in order to not make the paper even more difficult to read.
> > We could propose to include it in the Appendix if you see it fit, please let us know?
> >
> > ## Regarding the Meta-Learning Context:
> >
> > We appreciate you clarifying your perspective on the similarities between the meta-RG algorithm and the common RG algorithm.
> >
> > Firstly, you make a good observation that the training stimulus sampling does not depend on parameter updates during training.
> > In fact, the algorithm modification that you propose makes the pseudo-code closer to our actual implementation, but we were not sure which version would be easiest to read through, and initially chose the other version...
> > We are updating the algorithm to reflect your preference, assuming it will be more widespread than ours.
> >
> > On the otherhand, you state that 'the two algorithms (Algorithm 4 and 5) can share the rest of the process in an episode', which makes us realise that we failed to emphasise how the position of the update involves access to different information and therefore the two algorithms cannot share the rest of the process:
> > Indeed, in the meta-learning context, the update is over the whole episode, thus involving both the supporting (training-like) phase, and the querying (testing-like) phase, because training for CLBs requires it, as opposed to training for CBs [1,2].
> >
> > We are adding mention of the dependencies of the update in each algorithms to try to address your feedback.
> >
> > Thank you again for your time and feedback.
> > Please let us know if there are any other ways we could improve your rating of the paper.
> > We greatly appreciate you taking the time to ensure quality publications by engaging in this discussion.

---

> > > ### Author Response · Authors · 2023-11-21
> > > **Response (3/3) : References for Meta-Learning Context**
> > >
> > > [1] : Lake, Brenden M. "Compositional generalization through meta sequence-to-sequence learning." _Advances in neural information processing systems_ 32 (2019).
> > >
> > > [2] : Lake, Brenden M., and Marco Baroni. "Human-like systematic generalization through a meta-learning neural network." _Nature_ (2023): 1-7.

---

### Official Review · Reviewer_3drE · 2023-10-31

**Soundness:** 2 fair
**Presentation:** 2 fair
**Contribution:** 3 good
**Rating:** 5
**Confidence:** 2

**Summary:**

The paper proposes a referential game benchmark to investigate the agent's ability to solve a domain-agnostic binding problem and exhibit compositional learning behaviors.

**Strengths:**

+ Originality:

    The proposed Symbolic Continuous Stimulus (SCS) and the meta-referential games benchmark built upon it are novel and interesting.

+ Significance:

    Probing and investigating the compositional learning behaviors are important for various machine learning communities.

**Weaknesses:**

- Quality & Clarity:

    i) I am a bit confused about the claim that the proposed SCS is *shape invariant*. What does this specifically mean in the context of this paper? Would be great if the authors can give a clear definition of this property.


   ii) Can the authors provide more insights and explanations about why SCS is a domain-agnostic representation?


   ii) What is the architecture used for the Recall task experiment in appendix C.1? Is it possible that the performance gap is caused by the choice of implementation of the agents? My concern is whether the proposed SCS is universally more effective than OHE in terms of BP, regardless of the network architectures. Is there any theoretical evidence of this claim?

  iv) How does the shape invariance property of the SCS representation translate into the meta-referential games?

   v) The description of the meta-referential games is a bit abstract to me. It's also unclear to me how the compositionality is examined through the games. It would be great if the authors can provide an algorithm table to summarize the game procedure and show some game instances to facilitate understanding.

**Questions:**

See the weaknesses section.

---

> ### Author Response · Authors · 2023-11-13
> **Rebuttal (1/3)**
>
> Thank you for your review, we address your feedback and questions below:
>
> ## Question 1 and 4: What does the SCS being shape invariant specifically mean in the context of this paper and the meta-referential games?
>
> In this paper, we present some symbolic stimuli to some learning AI agent.
> The common approach when dealing with symbolic stimuli is to use the one-/multi-hot encoded representation. Since we consider compositionality, we ought to compare to the multi-hot encoded representation which supports compositionality: for instance, when dealing with a semantic space with $N_dim=2$ dimensions, where each dimension represent a different attribute for our semantic space, then each attribute/factor dimension allows instantiation of different values. For instance, on attribute/factor dimension $1$, we might have 3 possible values (meaning that $d(1)=3$), and on the second one, we might have 2 possible values (i.e. $d(2)=2$).
> Then, let us sample a stimulus out of this semantic space: for instance $s=(2,1)$, which instantiate value $2$ on attribute/factor dimension $1$, and value $1$ on attribute/factor dimension $2$.
> Typically, using the multi-hot encoded representation, we would represent this stimulus using $5$ binary digits (i.e. $d(1)+d(2)$), as follows : $(0 1 0 1 0)$.
> In this representation, the first three binary digits are used for the encoding of the value of the attribute/factor dimension $1$ (since it has $3$ different values that could be instantiated), while the last two binary digits encode the value instantiated on attribute/factor dimension $2$.
>
> On the otherhand, with the SCS representation, we would be using a vector with $2$ dimensions, one for each attribute/factor dimension of the sampled stimulus.
> Without the need to enter into the details of this vector, we bring the reader attention onto the fact that the shape of the SCS representation does not depend on the number of possible values that can be instantiated on each attribute/factor dimensions (i.e. the $d(i)$'s), on the contrary to the OHE/MHE representation.
> This means that if I sample a stimulus from a similar $N_{dim}=2$-semantic space, but with different number of possible values for each attribute/factor dimensions (i.e. different $(d(i))_{i\in [1,2]}$), then the shape of the SCS representation will not change, it will still be a $2$-dimensional vector. This invariance with respect to the structure of the semantic space is what we refer to as the shape invariant property.
>
> In this paper, at each meta-referential game, a different structure for the semantic space is sampled, but the same AI agents are used, so it is important that the stimuli representation remains compatible with our AI agent, therefore it is best if it does not change from one meta-referential game to another.
>
> ### What about using an OHE/MHE representation based on the maximal number of values that any attribute/factor dimension may have throughout all the possible meta-referential game that we want our agents to play, i.e. $V_{max}$ ?
>
> While this would add shape invariance to the MHE representations of all stimuli coming from any semantic space's structure that we would want to consider, it would come with two limitations:
>
> Firstly, using the OHE/MHE representation as such would still imply that we would need to build into the AI agent a limitation on its input format, whereas the SCS representation theoretically allows us to input any stimulus coming from any symbolic space's structure. By using the term theoretically here, we mean to nuance that while we can feed such stimulus to the SCS-expecting AI agent, it remains to be seen whether the AI agent can operate on it, e.g. if we have only trained the AI agent with $V_{max}=5$ then if we suddenly feed it a stimulus that instantiates a 10-th value on a given attribute/factor dimension, then this is requiring the AI agent to perform out-of-distribution generalisation.
> This is a problem that we hope to tackle in a future work, and thus the SCS representation is necessary to tackle it.
>
> Secondly, the OHE/MHE representation depends on some specific binary digits that directly represent the value instantiated on each attribute/factor dimension. In other words, even with the proposed $V_{max}$-related trick, the OHE/MHE representation still leaks some information to the AI agent about the structure of the semantic space it deals with. For instance, from the observation of a single stimulus whose OHE/MHE representation has its 3rd binary digit toggle to 1, it provides the information that there is at least $3$ possible values on attribute/factor dimension $1$.
> This kind of information leakage will prevent instantiation of a proper Binding Problem and allow emergence of a cheating language as we described in Appendix C.

---

> ### Author Response · Authors · 2023-11-13
> **Rebuttal (2/3)**
>
> ## Question 2: Why is SCS a domain-agnostic representation?
>
> By the word 'domain', we refer to the different modalities that stimuli may come from, e.g. vision (for the visual domain), text, sound, etc...
> Thus, by calling the SCS representation domain-agnostic, we imply that there is no longer any tie to any specific domain and therefore the data/stimulus may come from any domain.
>
> VAEs have been shown to being able to encode/embed/map data coming from any domain into their latent space. Therefore, we acknowledge their latent space to be domain-agnostic. And, since   the SCS representation mimics the latent space of an idealised VAE (with Gaussian kernels), we argue that the SCS representation is domain-agnostic.
>
> ## Question 3a: What is the architecture of the agent in Experiment C.1 ?
>
> The agents are the same baseline RL agents used in the main experiments presented in Section 4 (third line of the paragraph starting with 'Evaluation.').
> In the case of OHE/MHE representation, we adapt the dimension of the input fully-connected layer accordingly though.
> We can add details in the appendix if you see it necessary, please let us know?
> We thought that our open-sourcing of the code was sufficient.
>
> ## Question 3b : Is it possible that the performance gap is caused by the choice of implementation of the agents?
>
> We cannot think of any other implementation choice for the MDP instantiated in the Recall task, but please let us know if you think that something specific ought to be tried.
>
> ## Question 3c: SCS being universally more efficient than OHE in terms of BP?
>
> Could you clarify further your question, please, as we are confused about what you mean?
> Indeed, the experiment show worse asymptotic performance on the Recall task when using SCS, compared to using OHE.
> Thus we are confused about what you refer to when calling the SCS being more efficient that the OHE.
>
> ## Question 5a: Addition of an algorithm table to better detail the meta-referential game
>
> Thank you for your suggestion, Reviewer CPjL also mentions it, we will add it to the appendix and attempt to draw a comparison with the normal referential game, sitting within a supervised learning loop, versus the meta-learning loop in which the meta-referential game sits.
>
> ## Question 5b: Details about how the compositionality is examined:
>
> Thank you for your feedback, but could you clarify which compositionality examination is ambiguous, please? Is it the linguistic one or the behavioural one, and which part exactly?
>
> The behavioural compositionality, the one that is relevant in regards to performing CLBs, is examined via the ZSCT accuracy, which is the accuracy reported over the RG taking place during the querying/testing-phase of an RL episode.
>
> It is detailed in Section 3.2. We reframe the section by departing from the terms 'training' and 'testing' and replacing them with 'supporting' and 'querying' in order to further emphasise the few-shot, meta-learning framework, and remove any confusion about when are agents' parameters update taking place.
> Moreover, we are adding in the Meta-RG algorithm (Algorithm 5) details about the ZSCT accuracy measure during the querying/testing phase.
> Please let us know if this clarifies your doubts?
>
> As far as the linguistic compositionality is concerned, we detailed how it is measured in the 'Evaluation' paragraph of Section 4.1. We are adding a reference to the newly-added common RG algorithm (Algorithm 4) which contains details regarding the ZSCT accuracy and systematic train-test split of the dataset.  Please let us know whether this addresses your confusion, and/or whether there remains any part of it that you would like to see further detailed.

---

> > ### Author Response · Authors · 2023-11-13
> > **Rebuttal (3/3)**
> >
> > Once again, thank you for your time and your feedback.
> >
> > In summary, in order to address your feedback, we are adding algorithms detailing further the meta-referential game process in comparison to the common RG, and we add a figure that illustrates the SCS representation's shape invariance property.
> >
> > Please let us know if those answers to your questions (above) and proposed changes (cf. updated PDF) contribute to increasing your rating of the paper.
> > And, in any case, please let us know if you have any subsequent suggestion of changes for us to improve your rating.

---

> > > ### Comment · Reviewer_3drE · 2023-11-21
> > >
> > > Thanks for the detailed and point-to-point responses. I appreciate the efforts that authors made to address my questions/concerns. Apologies I have very little background in the area of referential games and should have probably opted out. However I did read the text twice with interest. I am still not very clear about how the choice of SCS representation leads to the compositional learning behavior, and would therefore keep my score as it is. I would not feel uncomfortable if reviewers more familiar with this area consider this work a promising contribution and should be accepted though.

---

> > > > ### Author Response · Authors · 2023-11-21
> > > > **Response**
> > > >
> > > > Thank you for your reply and your effort towards understanding.
> > > > We would like to keep the conversation going in order to try to address a possible misunderstanding that is cued by your sentence saying that it is not clear 'how the choice of SCS representation leads to the compositional learning behavior' (CLBs) :
> > > >
> > > > This sentence seems to imply that you hold the SCS representation as a feature that enables CLBs to be exhibited by the learning agents.
> > > > We are writing in the assumption that this is your understanding and we mean to clarify that there might lie a misunderstanding there :
> > > > Choosing the SCS representation, instead of the OHE/MHE representation for instance, is not a design choice that is meant to further the learning agent's ability to perform CLBs.
> > > > Rather, it is a design choice for the sake of the benchmarking : using the SCS representation is solely meant to enable the benchmark to instantiate at each episode a never-before-encountered (domain-agnostic) Binding Problem (BP), for the agent to be evaluated against.
> > > >
> > > > Note that we tried to cristallize this aspect in the paper in the following sentence, situated in the first main-text paragraph of Page 5 : "While the SCS representation is inspired by vectors sampled from VAE’s latent spaces, this representation is not learned and is not aimed to help the agent performing its task. It is solely meant to make it possible to define a distribution over infinitely many semantic/symbolic spaces, while instantiating a BP for the agent to resolve."
> > > >
> > > > Please let us know if any of this helps clarifying any possible misunderstanding.
> > > > And if so, please let us know whether you see any possible change that could help make the paper more readable around that aspect of the work, and thus possibly increase your rating of it.

---

### Meta-Review · Area_Chair_XXNh · 2023-12-08

**Metareview:**

This paper presents an interesting benchmark to assess compositional learning behaviors in artificial agents, and in particular the “receptivity” and “constructivity” aspects as defined in Santoro et al, 2021, by proposing a meta-learning extension of referential games, called meta-referential games. They evaluate R2D2 RL multi-agents and perform empirical analyses.

Reviewers expressed concerns about lack of novelty compared to common referential games (qdfE), the significance and interpretability of their results (Hoy7), and all indicated that clarity and presentation could be improved. The authors responded at length during the discussion, and all reviewers engaged.

Reviewers asked a lot of clarifying questions but the authors' responses didn’t seem to really clear things up, indicating the presentation could use some work to be fit for a more general ML audience. The two reviewers who scored 6 never expressed strong conviction to accept, even after extensive engagement with the authors. For instance, I quote from reviewer Hjoy7’s last response:
“I think the considerations brought up are important and do a good job of situating the work, but from an EC-centric perspective, I am skeptical of small-scale benchmarks' practical use.”

Having skimmed through the paper myself, I find myself agreeing with this viewpoint, and further believe they could have evaluated on more than 1 type of agent to showcase their benchmark. As there are much stronger papers in my batch, I believe this paper isn’t ready for publication at ICLR in its current form, but hope that the reviewers’ extensive comments are helpful for future versions of this work. The authors might also consider a benchmark-specific venue, such as NeurIPS's Datasets and Benchmarks track.

**Justification For Why Not Higher Score:**

Having read through the 2 reviews recommending borderline accept as well as the exchange with authors, I found no strong reasons to accept, as this was a relatively minor extension to common referential games, and evaluated on only 1 type of RL agent. The presentation and clarity was lacking, and so could use at least another round of major revisions and thus this paper is not publication-ready.

**Justification For Why Not Lower Score:**

N/A

---

### Decision · Program_Chairs · 2024-01-16

Reject